# Accurate graphene quantum Hall arrays for the new International System of Units

**Hans He** [1] ✉, **Karin Cedergren** [1], **Naveen Shetty**[2], **Samuel Lara-Avila** [2,3], **Sergey Kubatkin**[2], **Tobias Bergsten** [1] & **Gunnar Eklund**[1]

Graphene quantum Hall effect (QHE) resistance standards have the potential to provide superior realizations of three key units in the new International System of Units (SI): the ohm, the ampere, and the kilogram (Kibble Balance). However, these prospects require different resistance values than practically achievable in single graphene devices (~12.9 kΩ), and they need bias currents two orders of magnitude higher than typical breakdown currents $I_C \sim 100$ μA. Here we present experiments on quantization accuracy of a 236-element quantum Hall array (QHA), demonstrating $R_K/236 \approx 109$ Ω with 0.2 part-per-billion (nΩ/Ω) accuracy with $I_C \geq 5$ mA (~1 nΩ/Ω accuracy for $I_C = 8.5$ mA), using epitaxial graphene on silicon carbide (epigraphene). The array accuracy, comparable to the most precise universality tests of QHE, together with the scalability and reliability of this approach, pave the road for wider use of graphene in the new SI and beyond.

The 2019 redefinition of the International System of Units (SI) has fundamentally changed the world of electrical precision measurements, and the base units are now derived from seven exactly defined fundamental constants[1], such as Planck's constant $h$ and elementary charge $e$. The quantum Hall effect (QHE) is one cornerstone in the SI, and epigraphene QHE devices have already revolutionized practical resistance metrology[2–4], becoming the preferred embodiment of primary electrical resistance standards due to their robustness and relatively relaxed measurement conditions. The QHE in epigraphene provides an exact relationship between resistance and fundamental constants $R = R_K/(4(N+1/2))$, where $R_K = h/e^2 \approx 25.8$ kΩ (von Klitzing constant) and an integer $N \geq 0$. Epigraphene combines large Landau level spacing[5] with high energy loss rates, resulting in larger $I_C$ compared to conventional semiconductors[6,7]. The QHE can thus manifest at higher temperatures $T$, lower magnetic flux densities $B$, and higher bias currents $I$ compared to traditional systems where dissipation occurs easier[3,8]. Moreover, the large quantum capacitance of epigraphene leads to a $B$-dependent charge transfer from the substrate, resulting in the widest resistance plateau observed to date, extending to $B > 50$ T[9,10]. The $N = 0$ plateau is not only the most robust, but also the most well-quantized and is therefore preferred for precision metrology[2–4]. All of these epigraphene-specific virtues translate into

highly robust quantization over a wide parameter space[3] and greatly facilitates practical quantum resistance metrology.

The QHE is gaining more prominence in the new SI due to its elevation from practical to true realization of the ohm, and it will serve other roles beyond resistance calibration. One application is the electrical realization of the kilogram via the Kibble balance[11], which in a nutshell is an instrument which measures the weight of an object by balancing the gravitational force with a compensating electromagnetic force, defined using $h$ via the QHE and the ac Josephson effect. Voltage measurements use primary Josephson voltage standards[12], but current measurements rely on secondary artefact resistors which have to be separately calibrated against QHE. Direct integration of QHE standards in the Kibble balance could increase its performance, while also decreasing the complexity of the measurements, ultimately leading to reduced uncertainties. Such a feat would require a device with 100 Ω resistance and $I_C$ on the order of 10 mA[13]. Furthermore, if QHE devices with arbitrary resistance and high $I_C$ could be implemented, they could be combined with existing programmable Josephson array voltage standards to realize the quantum ampere over ranges far beyond current pumps[14], and without as high external amplification[15,16]. Due to their stability, QHAs are also desired for precision measurements of current in general[17]. Moreover, QHA devices with different resistances

[1]RISE Research Institutes of Sweden, Borås, Sweden. [2]Department of Microtechnology and Nanoscience, Chalmers University of Technology, Gothenburg, Sweden. [3]National Physical Laboratory, Teddington, UK. ✉e-mail: Hans.he@ri.se

are also useful for practical resistance metrology and will reduce uncertainties in calibrations of a wide range of resistance values since they allow for direct comparison measurement between primary quantum standards and secondary standards, shortening the calibration chain. However, a technological breakthrough is needed to enable the aforementioned applications, since a single graphene Hall bar (HB) can in practice only achieve $R = R_K/2$ and $I_C \sim 100\,\mu A$ at typical operating conditions[2–4,18].

Arrays provide an elegant way to achieve quantized resistances at arbitrary levels via series and parallel connections of individual HBs[19–25], while effectively increasing $I_C$ via parallel connections. These benefits have been recognized for decades, starting as early as 1993 with arrays for improved Wheatstone bridges[26–28]. The first reports on precision measurements for a modestly sized array was reported in 1999[29]. Many laboratories have since pursued arrays, typically with resistances between 100 Ω and 1 MΩ. A 100 Ω quantum standard is of special interest, since 100 Ω is commonly used as a stable transfer standard in resistance metrology and covers a wide range of useful resistances for precision measurements. However, large QHAs have not until now matched the performance achieved by single HB primary metrological standards in terms of precision, reliability, and reproducibility. One great challenge is the need for 100% device yield Any minor imperfection in any individual HB, be it improper quantization or poor contact resistance, will be detrimental to array accuracy. In practice, this implies that achieving sub part-per-billion accuracy requires that the combined effects of device homogeneity, contacts, wiring, and residual longitudinal resistance $R_{XX}$ should be less than 100 nΩ for a QHA with resistance of 100 Ω. Another unresolved issue is the measurement of vanishing longitudinal resistance $R_{XX} \sim 0$, an established quick test of resistance quantization and a way to asses QHE devices before precision measurements[8]. While $R_{XX}$ can be measured

for individual HBs one at a time, this approach is not feasible for large-scale arrays.

Here we present QHE measurements performed on a QHA device consisting of 236 individual epigraphene HBs. Reliable microfabrication is achieved using uniform doping via molecular dopants[30], and minimized influence from contact and lead resistances using multi-terminal connections[26] and superconducting leads[21,31]. Direct comparisons between two QHAs using high-precision measurements show no significant relative deviation of their resistances within 0.2 nΩ/Ω, which demonstrates a mutual agreement comparable to the best universality tests of QHE to date[3,27,32,33]. Our measurements are validated through additional comparisons between QHAs, a single epigraphene HB, and a secondary 100 Ω standard. These tests demonstrate that the QHA is truly quantized to a high degree of accuracy and precision. Furthermore, we propose that direct comparisons between two QHAs, based on established QHE universality tests[3,32,33], is a precise and reliable method to test the quantization for routine measurements, serving a similar purpose as measuring $R_{XX}$ and contact resistance for individual HBs. In the end, our measurements show that large arrays can meet the stringent criteria set by single HB metrological standards, and QHAs can be used as a primary standard which can exceed traditional single HB devices in terms of applications. The accuracy of our QHAs, combined with the scalability and reliability of this approach, pave the road for superior realizations of three key units in the modern SI: the ohm, the ampere, and the kilogram.

## Results
### Device design
The array contains 236 individual Hall bars (Fig. 1), divided between two subarrays (Array1 and Array2) connected in series, each with 118 Hall bars in parallel and a nominal resistance of $h/(236e^2) \approx 109\,\Omega$

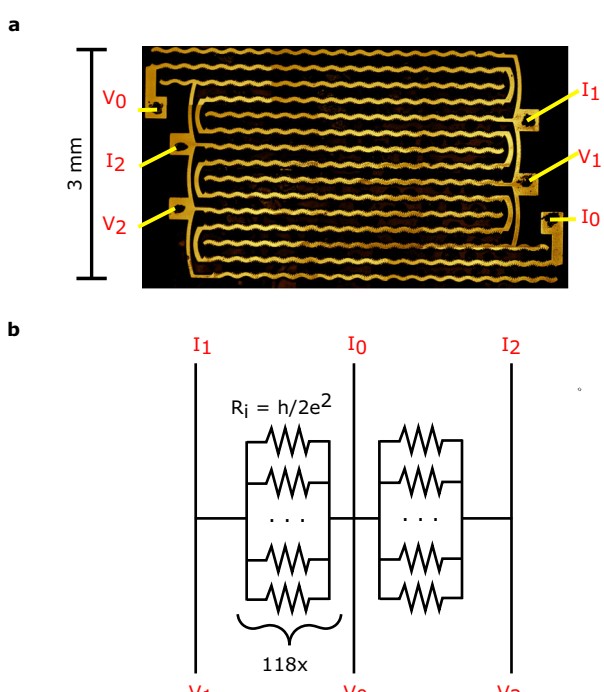

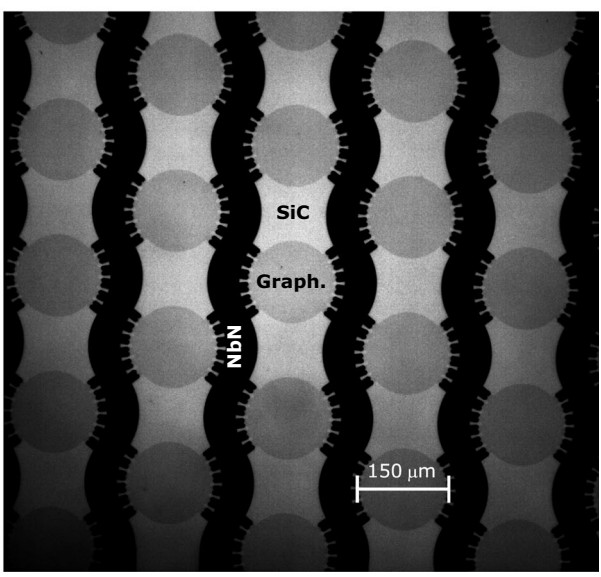

**Fig. 1 | Array design. a** Image is a false color composite micrograph of the whole array. It consists of two subarrays connected in series, each with 118 Hall bars in parallel for a total of 236 Hall bars. Subarray 1 and Subarray 2 are biased and measured using superconducting NbN leads connected to NbN pads (named $I_{0,1,2}$ or $V_{0,1,2}$). Subarray 1 can be measured by sending current between pads $I_1$ - $I_0$ and measuring voltage between $V_1$ and $V_0$. Subarray 2 can be addressed in a similar way using $I_2$–$I_0$ and $V_2$–$V_0$. **b** Simple schematic representation of the array. In quantum Hall regime,

each subarray can be represented by 118 parallel resistors, each with $R_i = h/(2e^2)$ resistance, here h is Planck's constant and e elementary charge. The full array consists of two subarrays connected in series. **c** Zoomed-in transmission mode micrograph of the individual circular epigraphene Hall bars (Graph.), which are connected in a simple two-probe configuration using NbN split-contacts with six prongs. The substrate (SiC) can also be seen. Each array element is topologically equivalent to a standard rectangular Hall bar wired in the multiple-connection configuration.

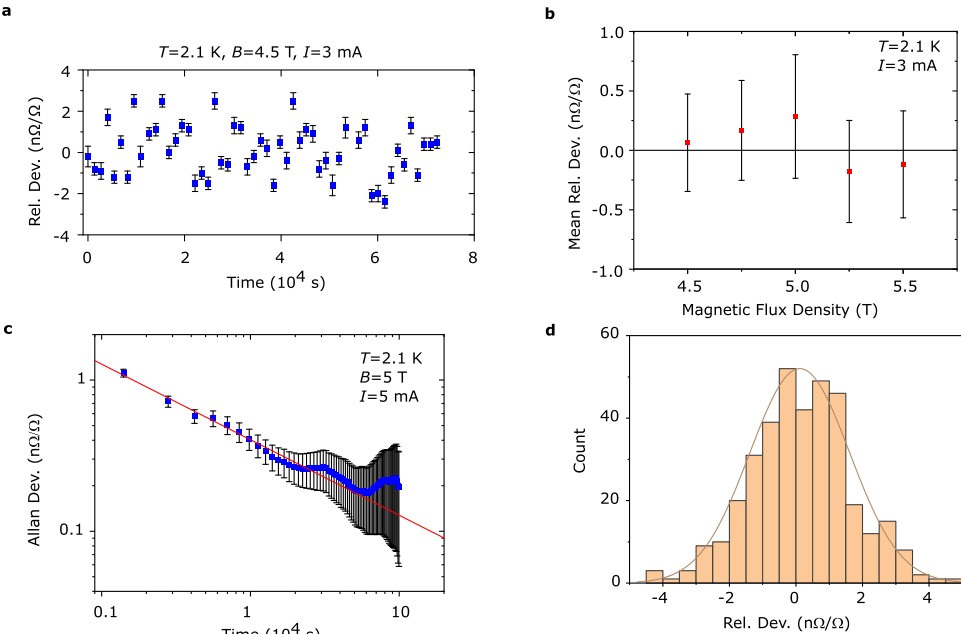

**Fig. 2 | Direct comparison measurements of epigraphene arrays. a** Precision cryogenic current comparator (CCC) measurements of subarray versus subarray comparison which shows their relative deviation at a certain magnetic flux density. This measurement consists of 53 CCC-readings, each around 20 min long. The error bars represent one standard deviation and are later used as weights in the final weighted mean, which in this case is 0.06 nΩ/Ω. **b** Precision measurements taken at different magnetic flux density show the mean relative deviation between the two subarrays. Each point is the weighted mean of ≥ 45 CCC-readings (like those in **a**), and the error bars represent one standard deviation of the mean, derived from Allan deviation at $10^4$ s (see **c**) for each measurement at a certain magnetic flux density. These measurements reveal that there is no significant deviation over the measured flux density range. **c** Allan deviation follows $1/\tau^{1/2}$ (red line), where τ is elapsed measurement time, which indicates that white noise dominates and limits the measurement uncertainty to 0.2 nΩ/Ω. At longer averaging times the Allan deviation no longer decreases due to the onset of other dominating sources such as 1/f noise and thermal voltage drift. The error bars are estimated relative errors (see Methods). **d** Histogram of the data which produced the means shown in **b**. Each count represents one 20 min long measurement series. The distribution is normal, and the solid line is a Gaussian fit which shows that the unweighted mean lies around 0.1 nΩ/Ω.

(whole array $R = h/(118e^2)$) at the $N = 0$ plateau. The Hall bars are circular in order to achieve symmetrical design with high packing density. To maximize $I_C$, the diameter was chosen to be 150 μm so that the minimum distance the QHE edge state needs to travel between the two source-drain contacts (approximately a quarter of the circumference, Fig. 1c) exceeds the equilibration length of the edge state, which is on the order of 100 μm at 5 T and 2 K[34]. In order to eliminate lead resistance, the contacts and interconnects were made from a superconducting film, following initial reports on primary standards using graphene arrays[21,22]. The arrays in this work use niobium nitride (NbN) (Supplementary Fig. 1), and the film was dimensioned to be at least 120 nm thick and 50 μm wide to support currents on the order of 10 mA at 5 T and 2 K[35]. The NbN is in direct contact with epigraphene, with a split contact design using six-pronged connections to minimize the contact resistance[26,31]. The NbN was deposited using sputtering, and was used in combination with a special fabrication method to create edge contacts to graphene[36] (see Methods). This method is key in ensuring reproducibly low contact resistances through the array. The carrier density was tuned using molecular doping[30], which reliably yields low charge disorder and proper quantization, and stability over years[18]. The array exists on the same chip together with individual Hall bars, and all measurements were performed in the same cryostat and using the same setup. The proximity of the devices minimizes external influences due to excess wiring, and the direct one-to-one ratio comparison between the subarrays further reduces uncertainty contributions and errors in the precision measurements. The subarrays were tested simultaneously by performing a direct comparison of their quantized resistances via a cryogenic current comparator (CCC) system, which is a well-established method to measure resistance ratios with the highest precision[2,3,18]. The CCC can

detect minute resistance deviations $\Delta$ from a nominal value of 100 Ω on the order of 10 nΩ (i.e., 0.1 nΩ/Ω)[37], and makes for the ultimate test of resistance quantization.

## Precision measurements of Hall bar array

Figure 2 contains the main results of the subarray comparison: the mean relative deviation of the direct subarray comparison demonstrating that the resistance of each subarray is the same within 0.2 nΩ/Ω. Figure 2a shows an example of a precision comparison measurement between the two subarrays. One full measurement set consists of multiple CCC-readings (≥ 45 readings), each taking 20 min and consisting of multiple current polarity shifts to compensate thermal voltages and short-term drift. Each data point in Fig. 2b is the weighted mean of such measurement sets, taken at various magnetic flux densities. The standard deviation of each reading (Fig. 2a) is used as a weight in the calculation of the final mean (see Methods). The error bars in Fig. 2b are the standard deviation of the mean for each measurement set, limited by Allan deviation at $10^4$ s. Note that all uncertainties in this work are stated with unity coverage factor ($k = 1$), unless otherwise specified. Allan deviation analysis is used to characterize the type of noise present in the measurements[38]. We observe a general decrease with elapsed measurement time $\tau$ as ∼ $1/\tau^{1/2}$ (Fig. 2c), indicating that white noise is the dominating type. This trend is broken at longer time scales since other sources of noise such as slow temperature drift and 1/f-type noise start to dominate, and more time averaging will not necessarily improve the final measurement uncertainty. In the limit of white noise, the minimum measured uncertainty for the standard deviation of the mean in our experiments is in practice 0.2 nΩ/Ω. A histogram (Fig. 2d) shows that the data used in the above analysis are normally distributed and further supports the notion that white noise dominates.

**Table 1 | Comparison between different quantum Hall arrays**

| Material | Hall bars | Nominal Res. ($R_K = h/e^2$) | Nominal Res. (kΩ) | Relative Deviation (nΩ/Ω) | Meas. Uncertainty k = 1 (nΩ/Ω) | Current (mA) | Temp. (K) | Magnetic Flux Dens. (T) |
|---|---|---|---|---|---|---|---|---|
| GaAs[23] | 100 | 1/200 | ~0.129 | 0.1 | 2 | 2 | 1.3 | 8.4–9 |
| GaAs[25] | 88 | 5075/131 | ~1000 | 20 | 8.5 | 0.001 | 1.5 | 9–10 |
| GaAs[29] | 10 | 5 | ~129 | 2.5 | 12.7 | 0.005 | 0.3–1 | 8–9 |
| Graphene[20] | 100 | 1/200 | ~0.129 | $10^7$ | $10^5$ | 0.1 | 2 | 7–9 |
| Graphene[21] | 6 | 2/6 | ~8.60 | 1.9 | 0.75 | 0.15 | 1.7 | 9 |
| Graphene[24] | 10 | 5 | ~129 | 10 | 20 | 0.5 | 4 | 6 |
| Graphene[22] | 13 | 1/26 | ~0.993 | 0.45 | 3 | 0.3 | 1.6 | 7.5–9 |
| Graphene (this work) | 236 | 1/236 | ~0.109 | 0.03 | 0.2 | 5 | 2.1 | 5 |
| Graphene (this work) | 236 | 1/236 | ~0.109 | 0.5 | 0.5 | 8.5 | 2.1 | 4.25 |

The table includes previous reports on quantum Hall arrays, including both traditional GaAs and modern graphene approaches. The table columns show the material, number of individual Hall bars, nominal array resistance, relative deviation from nominal value, measurement uncertainty, measurement current, temperature, and magnetic flux density range. The result from this work is included in the bottom two rows. There are two subarrays, each consisting of 118 individual Hall bars, resulting in 236 elements for the entire array.

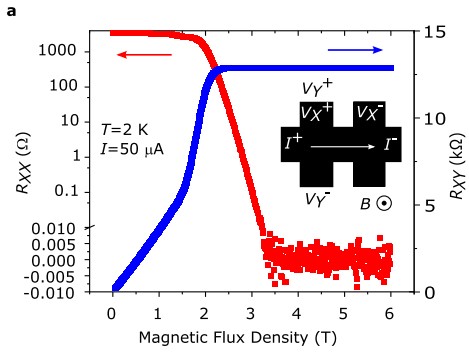

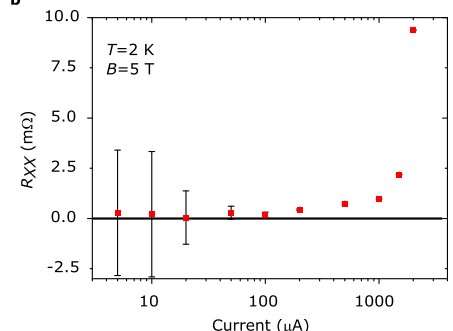

**Fig. 3 | Characterization of single Hall bar.** *a* A separate Hall bar with a normal rectangular geometry is used to measure longitudinal resistance $R_{XX}$ (red) and transverse resistance $R_{XY}$ (blue). The device is fully quantized for B > 3 T, and the longitudinal resistance vanishes below the noise level of -100 nV. From the low-flux density measurement of the transverse resistance, the carrier density is determined to be $n = 1.7 \times 10^{11} \text{cm}^{-2}$ and mobility is µ = 19,600 $\text{cm}^2 \text{V}^{-1}\text{s}^{-1}$. Since all quantum Hall devices are located on the same chip, this also provides an indirect measurement of

the array carrier density and mobility. The Hall bar schematic in the inset shows how resistances are measured. Current $I_X$ is sent between contacts named $I^+$ and $I^-$. $R_{XY}$ is measured using contacts named $V_Y^+$ and $V_Y^-$ as $R_{XY} = V_Y/I_X$, while $R_{XX}$ is measured between $V_X^+$ and $V_X^-$ as $R_{XX} = V_X/I_X$. **b** Breakdown current measured on the Hall bar reveals no significant increase in $R_{XX}$ up to 100 µA bias. Error bars represent one standard deviation.

The weighted mean of the mean relative deviations $\Delta_{\text{Array1-Array2}}$ at different flux densities in Fig. 2b reveals the level of quantization[3,21,33]. Using the respective standard deviation of the mean as the weights (see Methods), the resulting weighted mean relative deviation and standard deviation of the weighted mean is $\Delta_{\text{Array1-Array2}} = (0.033 \pm 0.082)$ nΩ/Ω. This degree of reproducibility in the quantization of such a large QHA is unprecedented for both GaAs[23] and graphene[20,22], and it is well below 1 nΩ/Ω which is the requirement for precision metrology[8]. The level of agreement between the resistance of our subarrays can reasonably only be attributed to exact resistance quantization. Especially because the subarrays, though nominally identical, are expected to have slightly different non-quantized resistance due to finite doping difference (Supplementary Fig. 2).

The result of precision measurements on arrays is comparable to the most precise comparisons of single graphene Hall bars versus GaAs in universality tests of QHE, which used the same methods and analysis, with a reported deviation of $\Delta_{\text{GaAs-Graphene}} = (-0.047 \pm 0.086)$ nΩ/Ω[33] and $\Delta_{\text{GaAs-Graphene}} = (-0.009 \pm 0.082)$ nΩ/Ω[3]. Another type of universality test was performed using a small 4 Hall bar GaAs array in a Wheatstone bridge setup, which tested the reproducibility of the QHE with an uncertainty down to 0.076 nΩ/Ω[27], and even 0.032 nΩ/Ω[39]. Note however that in all above cases, the experimentally motivated level of uncertainty from Allan deviation is at best around 0.1 nΩ/Ω[3,27], comparable to our value of 0.2 nΩ/Ω.

Table 1 shows a comparison between our results and other reports on arrays in literature.

**Comparison between array and single Hall bar**

To validate our subarray comparison measurements, we have also compared the subarrays to an on-chip single Hall bar. These measurements are crucial to verify the subarray quantization accuracy and to form a link between our measurements and traditional quantum Hall experiments[8]. The Hall bar was dimensioned to be 200 µm wide, comparable to an individual array element, so that their $I_C$ are similar. The Hall bar characterization (Fig. 3a) shows that its longitudinal resistance $R_{XX} = V_X/I_X$ vanishes into the noise level of -100 nV (limited by setup) above the quantizing flux density B = 3 T, same as for the array (Supplementary Fig. 1). Figure 3b shows the bias current dependence of $R_{XX}$ of the Hall bar has no significant change up to 100 µA, and the $I_C$ for the Hall bar is therefore around 100 µA. This also suggests that the $I_C$ of an individual array Hall element should be on a similar level. The mean residual $R_{XX}$ for bias currents 5–100 µA is $R_{XX} = (0.2 \pm 0.2)$ mΩ (k = 2), which approaches zero within the noise. A residual resistance of 0.1 mΩ could lead to a deviation of the quantized resistance $h/(2e^2)$ on the order of 3 nΩ/Ω[3], and would be easily identified in CCC-measurements. The contact resistances (same NbN split contacts as array) were measured under quantizing conditions using a standard 3-probe configuration[8] and were all < 2 Ω, including -1.5 Ω

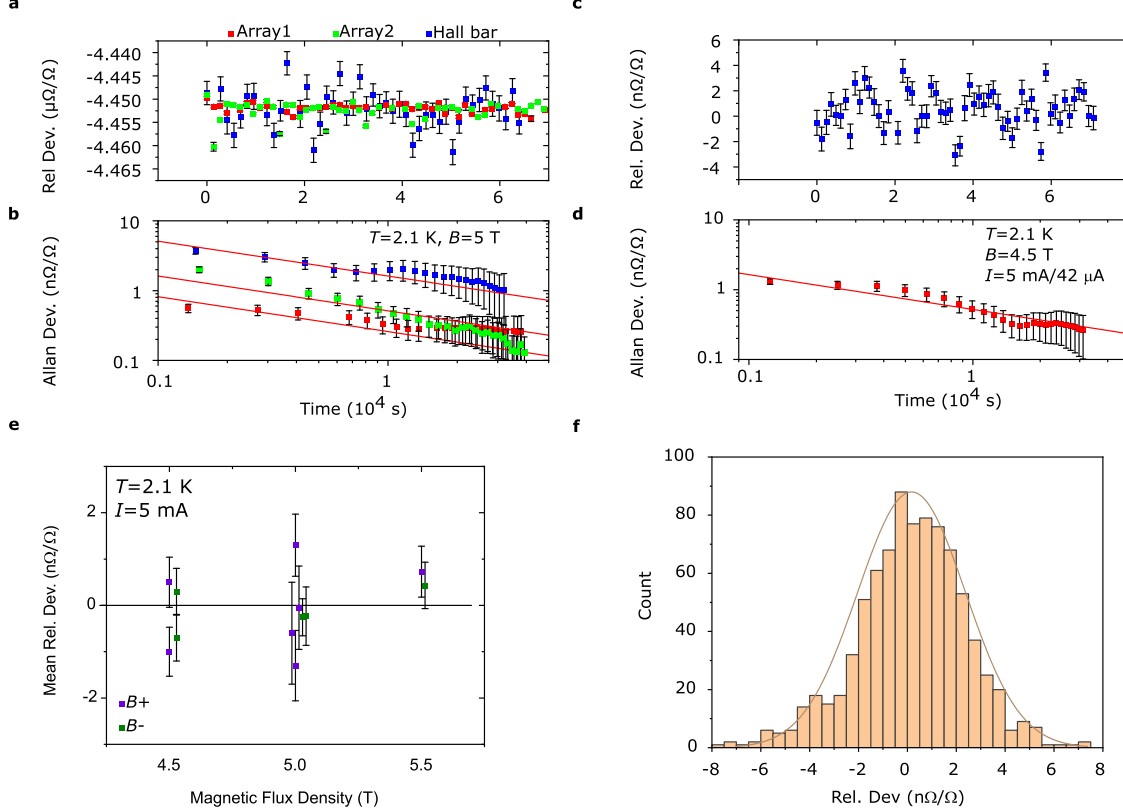

**Fig. 4 | Indirect and direct comparison between array and standard Hall bar.**
**a** Precision measurements of a 100 Ω resistance standard using both a standard Hall bar (blue) and two subarrays (Array1 red and Array2 green). The 100 Ω is biased with 3 mA and the Hall bar and subarray receive 23 μA and 2.75 mA respectively. The graph shows three different sets of cryogenic current comparator (CCC) readings of relative deviations, with one standard deviation error bars. **b** The corresponding Allan deviations, with estimated relative error (see Methods). The red line shows the corresponding Allan deviation for pure white noise. The mean relative deviations are $\Delta_{HB\text{-}100} = (-4.4521 \pm 0.0019)$ μΩ/Ω, $\Delta_{Array1\text{-}100} = (-4.4519 \pm 0.0003)$ μΩ/Ω and $\Delta_{Array2\text{-}100} = (-4.4522 \pm 0.0005)$ μΩ/Ω, with standard deviation of the mean taken from Allan at $10^4$ s. **c** Example of a direct comparison between Hall bar and one subarray. The top graph contains relative deviation data and error bars

representing one standard deviation. **d** The corresponding Allan deviation, with estimated error (see Methods). The standard deviation of the mean is limited by Allan to -0.2 nΩ/Ω. **e** Mean relative deviation for direct comparison between Hall bar and array, calculated from precision measurements like in **a**. The purple data represent positive flux density direction, while green represent negative flux density direction. The error bars represent one standard deviation of the mean, taken from Allan deviation at $10^4$ s (like **d**). The measurements were taken at three different flux density strengths (4.5, 5.0 and 5.5 T), but the data has been offset in the x-axis for clarity. **f** Histogram of the data which produced the means in **e**. Each count represents one 20 min long CCC-reading. The distribution is normal and centered around 0.17 nΩ/Ω.

wire resistance, which are well-below recommended levels[8]. In summary, the Hall bar passed all established tests for initial characterization of a quantum resistance standard.

Figure 4a,b shows the comparison between the Hall bar and a 100 Ω standard resistor, and each subarray versus the same 100 Ω standard. The 100 Ω standard is kept immersed in a temperature-controlled oil bath, with a well-recorded history and long-term stability. An indirect comparison between the Hall bar and subarrays using these data results in the deviation and combined uncertainty of $\Delta_{HB\text{-}100}$ - $\Delta_{Array1\text{-}100} = \Delta_{HB\text{-}Array1} = (-0.2 \pm 1.9)$ nΩ/Ω and $\Delta_{HB\text{-}Array2} = (0.1 \pm 2.0)$ nΩ/Ω. The combined uncertainty is dominated entirely by the Hall bar measurement, which is noisier because the resistance ratio of $h/2e^2$ compared to 100 Ω is far from unity, and therefore more sensitive to noise in the CCC-balance. The measurement noise is greatly reduced for the comparison between subarrays and the 100 Ω standard, and the indirect comparison between subarrays is $\Delta_{Array1\text{-}100}$ - $\Delta_{Array2\text{-}100} = \Delta_{Array1\text{-}Array2} = (0.3 \pm 0.6 \text{ nΩ/Ω})$, which is in good agreement with the direct array comparison.

To complete the set of comparison measurements, we also performed a direct comparison of the Hall bar and one subarray. Figure 4c shows one long CCC measurement, including Allan deviation (Fig. 4d). Figure 4e shows similar measurements taken at different magnetic flux densities and the histogram in Fig. 4f shows that the data is dominated

by white noise. Taking the weighted mean of all points (same as for data in Fig. 2b), the calculated mean deviation for the direct comparison is $\Delta_{HB\text{-}Array1} = (-0.04 \pm 0.2)$ nΩ/Ω, in good agreement with the direct subarray comparison. We have now demonstrated agreement between different combinations of direct and indirect comparisons between a quantized standard Hall bar, a 100 Ω standard, and the subarrays, and the measured deviations are all consistent with each other (Supplementary Fig. 3). These measurements cement the fact that the subarrays, and the Hall bar, are perfectly quantized with no discernable deviation from their nominal resistance values.

## Operation at high bias currents
Finally, we explored the performance limits of the arrays in terms of bias current, with the goal of determining when the QHE breaks down. The previous precision measurements (Fig. 4e) already show that a bias of at least 5 mA is possible without disturbing the sub-nΩ/Ω precision of the array. By increasing the bias current, we observe that deviations around 1 nΩ/Ω are possible at currents up to 10 mA and flux densities of 5 T (Fig. 5 and Supplementary Fig. 4). The quantization at elevated bias currents was tested by performing precision measurements at different flux densities (Fig. 5d). The apparent magnetic flux density dependence indicates that $I_C$ is at its limit for epigraphene (imperfect quantization), NbN contacts (resistive state), or a

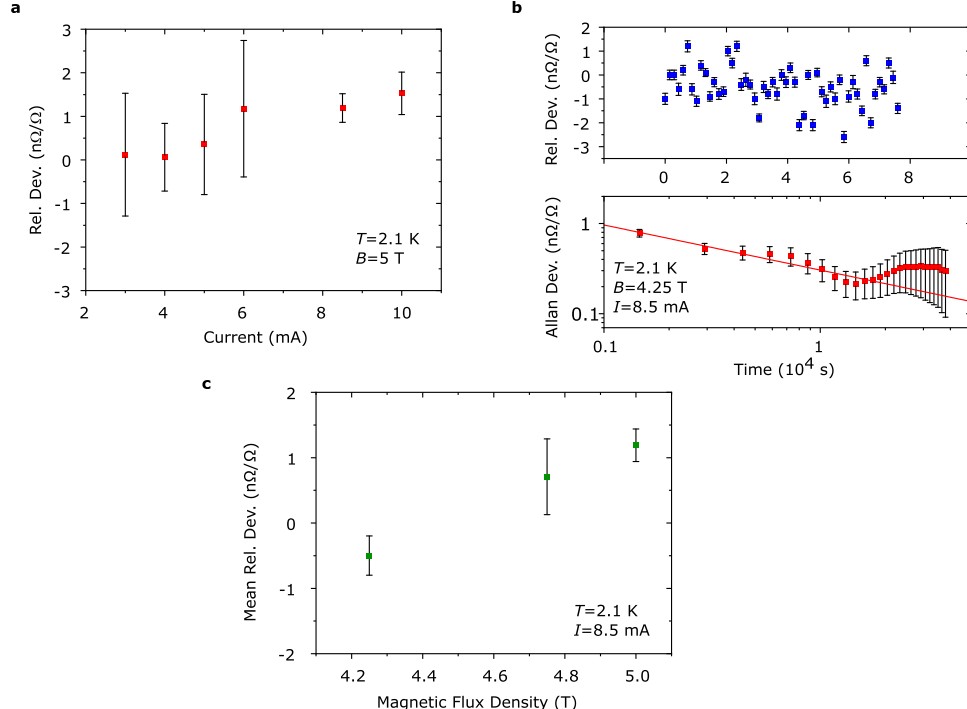

**Fig. 5 | High bias current measurements on arrays. a** Precision cryogenic current comparator (CCC) measurements using direct comparisons between subarrays shows no significant deviation until 8.5 mA. The data consists of the mean of 5 – 10 measurements each (20 min long) so there is no Allan deviation analysis, and the error bars are therefore one standard deviation. **b** Longer precision measurements showing the relative deviation, where each point is a 20 min long CCC-reading, with error bars of one standard deviation. **c** Corresponding Allan deviation, including estimated error (see Methods). The standard deviation of the mean is limited to 0.25 nΩ/Ω. The red line is a fit which shows the corresponding Allan deviation for pure white noise. **c** Mean relative deviation calculated from measurements like those in **b**. The error bars represent one standard deviation of the mean, taken from Allan deviation at $10^4$ s. There appears to be a significant deviation at 5 T, which disappears into the measurement uncertainty (k = 2) at lower flux densities.

combination of both, since $I_C$ can improve at lower flux densities for either[4,35]. The deviation at 8.5 mA is within 1 nΩ/Ω at lower magnetic flux densities < 5 T, which is acceptable for most practical metrological applications[8,18], including the Kibble balance[13]. Note that the fabrication techniques employed herein allow for further performance improvements. The observed $I_C$ is still far from any fundamental material limit and is simply restricted by the current device design. Since the NbN-leads can easily be made much larger (e.g., thicker film), what ultimately limits the QHA breakdown current is the single graphene Hall bar $I_C$, which can be much higher than the ~ 85 μA of current flowing through each individual array element at 10 mA. By tuning the carrier density to a higher value[30], an array with $I_C$ > 10 mA and good quantization should be achievable at 2 K and 5 T[18], and $I_C$ can be even higher under other operating conditions[3].

## Discussion

In summary, we have demonstrated that a graphene QHA consisting of 236 Hall bars is quantized with a precision of 0.2 nΩ/Ω, verified by traditional comparisons to a single Hall bar device. The highest precision quantization remained at large bias currents, up to at least 5 mA, with potential for operation at 8.5 mA and beyond. The reliable fabrication of such a precise array is dependent on key enabling technologies such as homogenous molecular doping and creation of low contact resistance superconducting leads on all array elements (the influence of contact resistance is estimated to be < 20 nΩ).

The proposed method of direct comparison of subarrays for routine measurements could be considered as an addition in future versions of practical metrological guidelines, which need to be revised given the new wave of quantum metrology devices based on epigraphene. The device design of future arrays should also be taken into

consideration. The subarray comparison measurements could be extended to allow for different ratio tests. For instance, a prospective 100 Ω array could be divided into 25 + 25 + 50 Ω parts. This would allow for non-unity ratio comparisons which serves as another quantization test and can also reveal potential errors like parallel leakages across the quantum Hall channel. Furthermore, one can also design two arrays with identical resistances, but using different amount of individual hall bars for each. This can be achieved via redundant parallel and series connections. Single hall bars elements in one array can be replaced by four elements, using two parallel connected sets of two serially connected Hall bars, without changing the total array resistance. This would mean that the current flowing through a single array Hall bar element would be different for the two arrays. A comparison measurement between the two would then be much more sensitive to potential quantization error, since if any of the Hall bars elements in either array deviate from perfect quantization, the different currents could lead to different resistance response. We foresee that these types of array-specific quantization tests will complement the existing single-hall bar tests in the future.

Embracing the use of array devices will allow for the QHE to be more intimately involved in the improvement of realizations of several key units, such as the ohm, ampere, and kilogram. We hope that our work will inspire further developments on this topic, and eventually lead to interlaboratory comparisons between different types of arrays, which is what is ultimately needed to establish graphene arrays as a primary resistance standard.

## Methods

The authors declare that all mentions of named products and companies are purely for reference and should not be taken as endorsements.

## Graphene growth

Epigraphene chips ($7 \times 7$ mm$^2$) were purchased from Graphensic AB. They were grown using thermal decomposition of silicon carbide[40] and had a monolayer coverage over 95%.

## Fabrication methods

For simplicity, the two subarrays were designed to consist only of parallel connections of individual Hall bars, with a single series connection between them. The number of parallel devices is 118 for each subarray, and this unusual resistance value of $h/(236e^2)$ was chosen because its ratio to 100 Ω is very close to 70/64, which is compatible with the winding ratios in the CCC[37]. The individual array Hall bar elements have a straight-forward minimalistic two-probe connection scheme in order to improve packing density, minimize complexity and increase device yield. Each hall bar element is contacted using two split contacts, each with six 15 μm wide prongs spaced 22 μm apart.

The devices were fabricated using standard electron beam lithography. Due to the nature of the chemical doping method, only poly(methyl methacrylate) (PMMA) resists are suitable to contact the surface of graphene. The first lithography step was to make the NbN-contacts. A special three layer resist structure was used[36] with PMMA directly on graphene (150 nm thick), followed by a copolymer poly(methylmethacrylate-co-methacrylic acid) (300 nm) and finally with AR-P 6200 (200 nm) on top. This combination ensures a resist profile where the middle layer has an undercut compared to the bottom and top layers, facilitating resist lift-off even for sputtered films. The exposure dose was tuned in such a way that the NbN film can be properly anchored to SiC, while still being in direct electrical contact to a graphene edge and forming an edge contact with low contact resistance[36]. After exposure and development in a mixture of isopropyl alcohol with 7% water, graphene underwent short reactive ion etching (RIE) with oxygen plasma (~30 s) to expose some of the SiC underneath. Then 120 nm of NbN was sputtered in a magnetron system, with the sample stage kept at room temperature. The sample was then immediately transferred to an electron beam evaporator to deposit a 20 nm protective layer of Pt to prevent degradation of the NbN-film. Lift-off process was performed using acetone. For the second and final lithography step, a single layer PMMA (150 nm) was used as a mask to define the Hall structures, and longer RIE in oxygen plasma (~1 min) was used to remove epigraphene.

After lithography, the sample was doped using chemical doping with F4TCNQ molecules[30]. This ensures a stable, homogenous, and controllable doping over the whole chip, with an expected charge inhomogeneity of doped graphene below $10^{10}$ cm$^{-2}$. Using thermal annealing at 160 °C to tune the carrier density, we aimed to achieve a carrier density on the order of $10^{11}$ cm$^{-2}$ which is suitable for quantum Hall measurements around 2 K and 5 T[4].

## Measurement setup

The devices were enclosed inside a dry TeslatronPT cryostat system, with a 12 T superconducting magnet and a base temperature of 1.5 K. The wiring consists of insulated copper leads with measured leakage resistance >25 TΩ. The influence of this leakage on a resistor of 100 Ω (subarrays are ~109 Ω) is entirely negligible, but for a normal quantum Hall resistance standard it can lead to an error in the comparison measurements on the order of 0.1 nΩ/Ω or more. However, this small deviation is usually within the noise level of the CCC-measurements.

All measurements were conducted at a temperature around 2 K, which was measured using a Cernox thermometer mounted next to the chip carrier. For the precise CCC-measurements, liquid helium was condensed inside the sample chamber and the sample was submerged in helium at 2.1 K, near the superfluid transition for the optimal temperature stability and maximal heat dissipation[41].

For regular measurements such as initial characterization, the samples were biased using a source (Keithley 6430 A) and measured using a nanovoltmeter (Keithley 2182 A). The measurement cables were twisted pairs copper leads with some shielding and no filtering, and the noise level was limited to ~100 nV.

The precision measurements were performed inside a CCC-system from Oxford Instruments. It can accurately compare two resistances by measuring their current ratio. For the comparison between Array1 and Array2 the winding ratios were set to Q = 64/64, for subarray versus 100 Ω standard Q = 70/64, Hall bar versus 100 Ω standard Q = 4130/32, and Hall bar versus 118× subarray Q = 3776/32. These ratios also determine the current ratio. The 100 Ω standard was always biased with 3 mA (limit due to heating), which automatically sets the current for the comparison QHE resistor (subarray or Hall bar) according to the resistance ratio. For comparison measurements between only Hall bar and arrays, various current levels were used to check the breakdown current.

## Data analysis

The CCC-system provides a measure of the resistance ratio $Q = R_B/R_A$ between two resistors A and B. This is then expressed as the relative deviation of test resistor B from its nominal value as referred to reference resistor A. The relative deviation can be expressed as $\Delta_{A-B} = (Q*R_A - R_{B,nominal})/R_{B,nominal}$. Here the value of $R_A$ is the reference value and is usually chosen to be a primary standard with a quantized resistance value like $h/2e^2$. $R_{B,nominal}$ is the nominal value of resistor B, and $\Delta_{A-B}$ describes how much the measured resistance of B (measured relative to $R_A$) deviates from its nominal resistance value.

Where applicable, the mean relative deviations are presented as the weighted means of CCC-readings using variance weights taken from each reading[42]. The weighted mean of $n$ samples of CCC-reading points $x_i$ with individual standard deviation $\sigma_i$ is:

$$\bar{x} = \frac{\sum_{i=1}^{n} x_i \sigma_i^{-2}}{\sum_{i=1}^{n} \sigma_i^{-2}} \quad (1)$$

With the experimental standard deviation of the weighted mean:

$$\hat{\sigma}_{\bar{x}}^2 = \frac{1}{\sum_{i=1}^{n} \sigma_i^{-2}} \frac{1}{(n-1)} \sum_{i=1}^{n} \frac{(x_i - \bar{x})^2}{\sigma_i^2} \quad (2)$$

Note that the standard deviation of the mean is sometimes used instead of standard deviation as the weight when calculating the mean of several means. Unless specified otherwise, the standard deviation of the mean is usually directly taken from Allan deviation analysis (in the region of white noise) instead of using the equation above. In fact, the $n^{-1/2}$ scaling of standard deviation of the mean is motivated only when white noise dominates.

The Allan deviation reported in this paper is the overlapping Allan deviation. For CCC-reading data $x_i$, in total $N$ samples, taken with $\tau_O$ time difference, and $n$ readings in a bin, the Allan variance at time $n\tau_O$ is calculated as:

$$\sigma_A^2(n\tau_0, N) = \frac{1}{2n^2\tau_0^2(N-2n)} \sum_{i=0}^{N-2n-1} (x_{i+2n} - 2x_{i+n} + x_i)^2 \quad (3)$$

The time difference $\tau_O$ is calculated as the average time difference between subsequent measurements. Each reading typically takes 20 min.

The relative error in for each point in the Allan deviation is estimated to be proportional to the inverse of the bin size[43] $n$:

$$err_{A\%} = \frac{1}{\sqrt{2(\frac{N}{n} - 1)}} \quad (4)$$

## Data availability

Relevant data supporting the key findings of this study are available within the article and the Supplementary Information file. All raw data generated during the current study are available from the corresponding authors upon request.

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

## Acknowledgements

This work was jointly supported by VINNOVA (Ref. 2020-04311 H.H. and 2021-04177 H.H.), the Swedish Foundation for Strategic Research (SSF) (Nos. GMT14-0077 S.K. and RMA15-0024 S.K.), 2D TECH VINNOVA competence Center (Ref. 2019-00068 S.L.), and Chalmers Excellence Initiative Nano S.L. This work was performed in part at Myfab Chalmers.

## Author contributions

K.C., T.B., H.H., and G.E. planned the experiments, H.H., N.S., S.L., and S.K. developed the fabrication processes, H.H. fabricated the graphene devices, H.H., K.C., T.B., and G.E. performed the electrical measurements and analyzed the data. H.H. and S.L. lead the writing process. All authors contributed to the writing process and reviewed the manuscript.

## Competing interests

The authors declare no competing interests.
