## [Peer Review File · Nature Communications]

Accurate graphene quantum Hall arrays for the new International System of UnitsREVIEWER COMMENTS

Reviewer #1 (Remarks to the Author):

The submission to Nature Communications describes an implementation of superconducting films combined with monolayer epitaxial graphene on SiC for metrology of the quantum Hall effect (QHE) standard for resistance. The authors utilize cryogenic measurements to show excellent quantization in an integrated device consisting of two sets of QHE devices, each consisting of 118 Hall bars in parallel. The bars are circular to minimize chip area, and feature multiple interconnections to eliminate parasitic resistance at the contacts. The manuscript focuses on two things, the low uncertainty obtained for the quantised resistance and several design features that improve the usefulness of the QHE device as a resistance standard. The authors emphasize the importance of these features, but do not explicitly credit the many earlier works in which similar advances were described. While the level of uncertainty achieved is impressive, claiming credit for these advances with a single implementation detracts from the work.

Technically, the explanation and treatment of the experimental process is complete and the results are fairly summarized.

P.1, Abstract and PP1, please define breakdown currents in terms of the effect on the measurement uncertainty and reference the guidelines for metrological QHE measurements (Delahaye and Jeckelmann, 2003, Ref. 22).

The development of QHE arrays has a long history which is omitted. The first published attempt for precision metrology was in 1999 (IEEE I&M, VOL. 48, NO. 2, APRIL 1999). There the resistance value of quantization was ten times the typical $i=2$ value, and the standard uncertainty was 12.9 parts in 10^9 . Many other QHE arrays have been developed over the past 20 years and their range of resistance includes the level of 100 ohms, as in these results. Thus, earlier studies for the 100 ohm level should be referenced, and directly compared with the new results, and some discussion about why the level of resistance near 100 ohms is of interest besides the increase in breakdown current would be helpful.

In PP2 the discussion of current measurement using the QHR and JJ, more recent references should be added for methods that utilize QHR arrays, for example in 2020 a paper appeared using "current-to-voltage conversion from a few nano amps to one microamp with the invariant nature of a $1\text{ M}\Omega$ QHRA and the programmable Josephson voltage standard (PJVS)." (Dong-Hun Chae et al 2020 Metrologia 57 025004). In the description of the Kibble balance, the term "weight" may be inappropriate since the weight is converted to mass by careful measurement of the gravitational acceleration near the experimental apparatus.

PP3, paragraph starting with "The use of arrays" is self-serving and unclear. Please revise the statement "However, QHAs have not until now truly met the stringent criteria required for a metrological standard in terms of precision and reliability." The term "metrological standard" is hard to define since many different levels of resistance exist, and the requirements of a standard change with each level. For example, arrays might be applied at the high resistance level above a megohm, where much less absolute precision is required. In recent literature at least three groups have demonstrated graphene QHR arrays and have published the results. The significance of these earlier works should be described in terms of investigating the use of superconducting interconnections, if present, the use of multiple device configurations for improved Wheatstone bridges, the development of quantum resistance for AC impedance standards, and the ability of array geometries to allow internal null comparisons as a check on quantization of the full array. Also, please add the reference below. Appl. Phys. Lett. 116, 093102 (2020); <https://doi.org/10.1063/1.5139965>

In the next paragraph, "a record-size" may be challenged. Does this refer to area or number of Hall bars? Please specify the chip area utilized. QHR arrays have been made using GaAs that have many more devices. See T. Oe, CPEM Conference Proceedings, <https://ieeexplore.ieee.org/iel7/8482236/8500784/08501182.pdf> Section: Device Design, PP1. Please give a reference for the value "109.376302794"

ohms and explain the significance of this value. Why are so many digits given? For the description "minimum distance between the two contacts" please refer to a figure and identify the contacts.

Section: Precision Measurements of Hall bar array; Please be clear in describing "sources of noise such as drift and 1/f-type..." since drift is attributed to some source (SQUID?) and typically is what constitutes 1/f type signal noise.

Section: Conclusion; the statement "The proposed method of direct comparison of subarrays..." implies that this is a new method not proposed before. Reference 19 makes use of this property of QHR arrays, as does the "Quantum Wheatstone bridge proposed by Delahaye and realized in GaAs and in graphene. Please note this and the recent usages of this method with graphene (references please): Martina Marzano et al 2020 Metrologia 57 015007; R. E. Elmquist et al "AC and DC Quantized Hall Array Resistance Standards," 2020 Conference on Precision Electromagnetic Measurements (CPEM), 2020, pp. 1-2, doi: 10.1109/CPEM49742.2020.9191854. The concept of an exact voltage balance in a Wheatstone Bridge using QHE standards is described in Delahaye's paper (Ref. 27).

Reviewer #2 (Remarks to the Author):

The Authors have reported an interesting study on graphene-based quantum Hall resistance standard. This work will certainly make an impact on the field of metrology, and their idea of using two graphene subarrays is interesting. Although this work warrants publication in some form, the Authors must address the following issues before I can recommend the publication of this paper in Nature Communications.

1. There are many grammatical errors in this manuscript. For example, all the physical quantities must be italicized. It really bothers me when I see something like $T=2K$ and $B=5T$. It should have been $T=2\text{ K}$ and $B=5\text{ T}$. These mistakes can be found in the figures as well. Shouldn't it be quantum Hall resistance standard instead of quantum Hall effect (QHE) resistance standard? It is the von Klitzing constant (not the Von Klitzing constant). $25,812.8074593045\ \Omega$ (15 significant figures)?

2. There are quite a few mistakes in the References. The page number of Ref. 36 is missing. The page numbers of Refs. 2, 20, and 29 must be wrong. This will almost certainly bug the Readers.

3. It is not clear to me how the Authors can compare subarray 1 (2) to a Hall bar device since the resistances of these two devices are not the same? I understand that they could tune the Q value yet could this introduce any certainties?

4. The Authors mentioned that "Any minor imperfection in any one individual Hall bar, be it improper quantization or poor contact resistance, will be detrimental to quantization accuracy." Could they tell the Readers how they are able to get around this issue, please?

5. The graphene array arrangement is the key to this nice work. However, to me, Fig. 1 (a) is not clear at all. Could they provide a schematic diagram in the supplementary information.

6. In Fig. S1 (a), the resistance appears to increase with decreasing temperature for $T < 11\text{ K}$. What is the reason for this?

7. The base temperature of their dry TeslatronPT cryostat is 1.5 K and most of the results are obtained at $T=2\text{ K}$. Performing the measurements at $T < 2\text{ K}$ could help?

8. Finally just some format issue. 5 T^{16} looks like Tesla to the 16th

power. They could have used 5 T (Ref. 16) instead.

Reviewer #3 (Remarks to the Author):

See attached file

Reviewer #4 (Remarks to the Author):

The authors describe the implementation of a new large-scale quantum Hall array resistance standard based on graphene on SiC and superconductive contacts. Experimental data display a reproducibility of 0.2 ppb and a large drive current. One possible application is the redefinition of the kilogram unit using a Kibble balance with an integrated resistance standard.

The manuscript is very interesting and timely and, to me, it looks suitable for publication. I just report down here a few minor points of improvement:

1) The authors mention that carrier density in their device is obtained by molecular doping, and measurements indicate $1.7 \times 10^{11} \text{ cm}^{-2}$ carrier density (a bit hidden in the caption to Fig.2). I could not find any indication on the expected homogeneity of the doping, Methods section just states the doping is "homogeneous". Any estimate of the homogeneity? Is it a relevant improvement parameter here? Given the accuracy, I guess all the graphene bars are well into the quantum regime as intended.

2) The device studied by the authors is interesting and somewhat non-conventional with respect to the ones typically used for QH standards. I believe that the term "Hall bar" might look a bit misleading to some readers: for instance, in a Hall bar, I would naively expect lateral contacts with independent wiring that allow proper 4 wire measurements, while here the device is rather built by putting many graphene conductors (each one with two "split contacts") connected in parallel. The four-wire measurement scheme is then realized by taking advantage of the zero-resistance superconductive leads... but basically this is a set of two-wire graphene conductors in parallel. So for instance R_{xx} cannot be measured not much because the array is large: given this contacting geometry (without independent lateral contacts) R_{xx} would be impossible to measure even if the array contained just a single graphene element!

I don't want to say this naming is "wrong": one can always think about the lateral contacts as shorted to the source and drain contacts of the Hall bar, or maybe just a "split-contact"; however, the authors could probably spend few words to communicate how this is not really a conventional Hall bar. I would anyway stress that the manuscript contains recent references 19 and 20 about these details, but that's a bit cryptic.

I also wonder if - beyond eliminating the lead resistance - the use of superconductive electrodes also contributes to minimizing the contact resistance to graphene of the split-drain and split-source. But this discussion would probably go out of the scope of the paper.

3) The paper contains some typos and broken sentences. I suggest a careful re-read. Few examples:

- caption to Fig.1. "... Allan deviation no longer decreases due to $1/f$ noise and drift (starts?) dominating. The error bars are estimated relative errors (?)."

- page 2 line 4, "they could using Ohm's law be combined...", maybe just a shuffling here.

Point-by-point reply

We would like to thank all the Reviewers for their in-depth questions. The Reviewers' comments are reproduced verbatim below and are written using normal font. The authors answers are given using both bold and italic font. We have also formatted each question and answer into its own bullet point for better readability. The changes in the revised manuscript are marked with yellow highlights.

Reviewer #1 (Remarks to the Author):

The submission to Nature Communications describes an implementation of superconducting films combined with monolayer epitaxial graphene on SiC for metrology of the quantum Hall effect (QHE) standard for resistance. The authors utilize cryogenic measurements to show excellent quantization in an integrated device consisting of two sets of QHE devices, each consisting of 118 Hall bars in parallel. The bars are circular to minimize chip area, and feature multiple interconnections to eliminate parasitic resistance at the contacts. The manuscript focuses on two things, the low uncertainty obtained for the quantised resistance and several design features that improve the usefulness of the QHE device as a resistance standard. The authors emphasize the importance of these features, but do not explicitly credit the many earlier works in which similar advances were described. While the level of uncertainty achieved is impressive, claiming credit for these advances with a single implementation detracts from the work. Technically, the explanation and treatment of the experimental process is complete and the results are fairly summarized.

We thank the reviewer for all the helpful comments. We believe we have implemented most of the Reviewer's suggestions. We highlight the newly introduced Table 1, which not only summarizes and credits earlier work, but also benchmarks our achievements versus earlier work including GaAs and Graphene devices. Below we provide a point by point response.

- P.1, Abstract and PP1, please define breakdown currents in terms of the effect on the measurement uncertainty and reference the guidelines for metrological QHE measurements (Delahaye and Jeckelmann, 2003, Ref. 22).

We have added these additional statements and references to the introductory paragraph on Page 1:

"These factors combined make it possible to have quantized resistance in graphene at higher temperatures, lower magnetic fields, and higher bias currents compared to traditional systems like GaAs. Outside of the parameter space set by the breakdown current, magnetic field, and temperature, the onset of dissipation destroys the precise quantization and has a detrimental effect on the measurement uncertainty"

- The development of QHE arrays has a long history which is omitted. The first published attempt for precision metrology was in 1999 (IEEE I&M, VOL. 48, NO. 2, APRIL 1999). There the resistance value of quantization was ten times the typical $i=2$ value, and the standard uncertainty was 12.9 parts in 10^9 . Many other QHE arrays have been developed over the past 20 years and their range of resistance includes the level of 100 ohms, as in these results. Thus, earlier studies for the 100 ohm level should be referenced, and directly compared with the new results, and some discussion about why the level of resistance near 100 ohms is of interest besides the increase in breakdown current would be helpful.

We have added a more thorough explanation of the history of QHE which will be of interest for a general audience unfamiliar with the field. A new section on Page 2 has been added:

“The benefits of array devices have been recognized for decades, with work towards array designs starting as early as 1993. They could be used to not only achieve different resistances, but could also be useful in improved Whetstone bridges. The first reports on precision measurements for a modestly sized array was reported in 1999. Since then, many laboratories around the globe have continued to develop arrays, typically with resistances in the range between 100 Ω to 1 M Ω . The 100 Ω is of special interest, since a 100 Ω secondary standard is commonly used as a stable transfer standard in resistance metrology and cover a wide range of practically useful resistance values for precision comparison measurements. However, large QHAs have not until now truly met the stringent criteria set by single Hall bar primary metrological standards in terms of precision, reliability, and reproducibility. “

We have also added extra clarification on the contact design (Page 2):

“Here we present quantum Hall measurements performed on a QHA device with a record number of 236 individual epigraphene Hall bars. Reliable microfabrication and uniform doping is achieved using a molecular dopant technique. The influence of contact resistance and leads can be reduced by using multiterminal connections, and recently it has been shown that superconducting leads can improve things further”

In order to complement the existing references in the main text, we have also added explicit comparisons between our works and many other previous arrays, see newly added Table 1.

- In PP2 the discussion of current measurement using the QHR and JJ, more recent references should be added for methods that utilize QHR arrays, for example in 2020 a paper appeared using “current-to-voltage conversion from a few nano amps to one microamp with the invariant nature of a 1 MOhm QHRA and the programmable Josephson voltage standard (PJVS).” (Dong-Hun Chae et al 2020 Metrologia 57 025004).

This reference has been added, along with the sentence (Page 2):

“Due to their stability, QHAs are also desired for precision measurements of current in general”

- In the description of the Kibble balance, the term “weight” may be inappropriate since the weight is converted to mass by careful measurement of the gravitational acceleration near the experimental apparatus.

We have clarified this section (Page 2) by changing it to:

“...gravitational force, which is later converted to mass using the measured local gravitational acceleration”

- PP3, paragraph starting with “The use of arrays” is self-serving and unclear. Please revise the statement “However, QHAs have not until now truly met the stringent criteria required for a metrological standard in terms of precision and reliability.” The term “metrological standard” is hard to define since many different levels of resistance exist, and the requirements of a standard change with each level. For example, arrays might be applied at the high resistance level above a megohm, where much less absolute precision is required. In recent literature at least three groups have demonstrated graphene QHR arrays and have published the results. The significance of these earlier works should be described in terms of investigating the use of superconducting interconnections, if present, the use of multiple device configurations for improved Wheatstone bridges, the development of quantum resistance for AC impedance standards, and the ability of array geometries to allow internal null comparisons as a check on quantization of the full array. Also, please add the reference below. Appl. Phys. Lett. 116, 093102 (2020); <https://doi.org/10.1063/1.5139965>

We have clarified this section by adding some additional history of arrays and their usefulness on Page 2, along with explicit mention of the development of superconducting leads:

“The benefits of array devices have been recognized for decades, with work towards array designs starting as early as 1993. They could be used to not only achieve different resistances, but could also be useful in improved Whetstone bridges. The first reports on precision measurements for a modestly sized array was reported in 1999. Since then, many laboratories around the globe have continued to develop arrays, typically with resistances in the range between 100 Ω to 1 M Ω . A 100 Ω quantum standard is of special interest, since a 100 Ω secondary standard is commonly used as a stable transfer standard in resistance metrology and cover a wide range of practically useful resistance values. However, large QHAs have not until now truly met the stringent criteria set by single Hall bar primary metrological standards in terms of precision, reliability and reproducibility.”

We have also added a comparison in Table 1 which includes the suggested reference and many others.

Regarding standards, we mean that arrays have yet to meet the standards set by single Hall bar devices, which currently are the primary realization of the resistance unit. Our goal is to make arrays fulfill the same stringent requirements as single Hall bar devices, so that they too can be a primary standard, and match or indeed surpass single Hall bar devices in terms of performance and usefulness. This includes aspects such as precision and reliability/reproducibility of the quantized resistance. We believe our work in this paper demonstrates that we can achieve this.

We have altered and added the following sentences on Page 2:

“However, QHAs have not until now truly met the stringent criteria set by single Hall bar primary metrological standards in terms of precision, reliability and reproducibility”

And Page 3:

“In the end, our measurements show that the arrays can meet the stringent criteria set by single Hall bar metrological standards, and QHAs can be used as a primary standard which can exceed traditional single device standards in terms of applications”

We have also added one additional reference with deal with universality test of QHE using a small array in a whetstone bridge, which presents one alternative way to check the reproducibility of the quantization of the whole array.

- In the next paragraph, “a record-size” may be challenged. Does this refer to area or number of Hall bars? Please specify the chip area utilized. QHR arrays have been made using GaAs that have many more devices. See T. Oe, CPEM Conference Proceedings, <https://ieeexplore.ieee.org/iel7/8482236/8500784/08501182.pdf>

We have clarified that our statement on “record-size” refers to number of individual graphene hall elements, which is directly related to the complexity of the array device. The focus is on graphene since it has become the premier material for quantum resistance standards. We want to emphasize that graphene is still a relatively new material compared to GaAs, and the production methods are not as mature yet. Our large array, especially combined with the record-high precision/reproducibility for its size, is a significant step forward for scaling up graphene electronics. The addition of a new table which contains explicit comparisons between arrays will also aid in highlighting our results. Note that we could only identify one larger GaAs array with 266 array elements (ours is 236), however it did not have any precision measurements and it is therefore not included in any comparison. Furthermore, the provided reference by T. Oe et al. contains only a design for a massive array above 700 elements but without any experiments, and therefore we do not include it either.

The section on Page 2 now reads:

“Here we present quantum Hall measurements performed on a QHA device with a record number of 236 individual epigraphene Hall bars.”

We have also added explicit comparison between various report on arrays in Table 1.

- Section: Device Design, PP1. Please give a reference for the value “109.376302794” ohms and explain the significance of this value. Why are so many digits given?

The nominal resistance of $h/236e^2 = 109.376302794...$ by definition in the new SI. The value is the nominal value for the array, assuming perfect contacts and quantization etc. The specific value was chosen because it is simple to achieve using only parallel connections, and it is supported by the winding ratios in our CCC (described in methods). However, to avoid added confusion we have changed the main text to $h/236e^2 \approx 109$ Ohm

- For the description “minimum distance between the two contacts” please refer to a figure and identify the contacts.

We have added additional information, an updated plot with a schematic, and a zoomed in detailed description of a single Hall bar element. It is the new “Figure 1” in the main text.

We are referring to the two source-drain contacts, and that the distance the quantum Hall edge state needs to travel between them is above 100 μm . We have added the sentence on Page 3:

“To maximize I_c , the diameter was chosen to be 150 μm so that the minimum distance the QHE edge state needs to travel between the two source-drain contacts (approximately a quarter of the circumference, Fig. 1b) exceeds the equilibration length of the edge state...”

- Section: Precision Measurements of Hall bar array; Please be clear in describing “sources of noise such as drift and 1/f-type...” since drift is attributed to some source (SQUID?) and typically is what constitutes 1/f type signal noise.

Noise sources such 1/f, or flicker noise, can be seen in Allan deviation as a flat plateau (no time dependence). This could be due to fluctuations in the SQUID. By drift we refer to slower processes, for instance temperature drift affecting thermal voltages and/or transfer standard TIN 100 Ohm (if applicable). These are things which fast current polarity shifts cannot eliminate fully. We have altered the sentence on Page 4 to read:

“This trend is broken at longer time scales since other sources of noise such as slow thermal drift and 1/f-type noise start to dominate, and more time averaging will not necessarily improve the final measurement uncertainty.”

- Section: Conclusion; the statement “The proposed method of direct comparison of subarrays...” implies that this is a new method not proposed before. Reference 19 makes use of this property of QHR arrays, as does the “Quantum Wheatstone bridge proposed by Delahaye and realized in GaAs and in graphene. Please note this and the recent usages of this method with graphene (references please): Martina Marzano et al 2020 Metrologia 57 015007; R. E. Elmquist et al "AC and DC Quantized Hall Array Resistance Standards," 2020 Conference on Precision Electromagnetic Measurements (CPEM), 2020, pp. 1-2, doi: 10.1109/CPEM49742.2020.9191854. The concept of an exact voltage balance in a Wheatstone Bridge using QHE standards is described in Delahaye’s paper (Ref. 27).

We have added additional clarifying statements to stress that the principle of the subarray comparison method is based on other comparison methods.

We simply propose that the direct subarray comparison should be used as an alternative to the standard single-device tests R_{xx} and contact resistance, for relatively quick characterization of arrays. These quick tests for quantization, useful for routine measurements, have not yet been established for arrays and remains one of the perceived drawbacks in the community. We demonstrate through thorough measurements that subarray comparison measurements can fulfil this role for large graphene arrays, and with an unprecedented degree of precision. Page 2 now contains the following:

“We propose that the direct comparison between two epigraphene QHAs, based on established QHE universality tests between GaAs and graphene, is the best method to quickly verify the quantization for routine measurements, serving the same purpose as measuring R_{xx} and checking contact resistance in individual Hall bars. In the end, our

measurements show that the arrays can meet the stringent criteria set by single Hall bar metrological standards, and QHAs can be used as a primary standard which can exceed traditional single device standards in terms of applications.”

Regarding Whetstone bridges, we have included mention of this in the main text. It is one option to compare 4 subarrays in a whetstone bridge (Schopfer et al., DOI:10.1063/1.2776371, now referenced in main text). However, our laboratory setup is most suited to the precise direct comparison between two subarrays using a CCC, which is why we don't focus on bridges. Schopfer et al. demonstrate their bridge measurements using a CCC as a null detector, which is one promising option for future investigations which can match, or even exceed, the precision we've demonstrated. We've added the reference by Marzano et al., but the suggested paper by Elmquist et al. appears to lack experimental measurements and therefore we do not include it.

Reviewer #2 (Remarks to the Author):

The Authors have reported an interesting study on graphene-based quantum Hall resistance standard. This work will certainly make an impact on the field of metrology, and their idea of using two graphene subarrays is interesting. Although this work warrants publication in some form, the Authors must address the following issues before I can recommend the publication of this paper in Nature Communications.

- 1. There are many grammatical errors in this manuscript. For example, all the physical quantities must be italicized. It really bothers me when I see something like $T=2\text{K}$ and $B=5\text{T}$. It should have been $T= 2 \text{ K}$ and $B= 5 \text{ T}$. These mistakes can be found in the figures as well. Shouldn't it be quantum Hall resistance standard instead of quantum Hall effect (QHE) resistance standard? It is the von Klitzing constant (not the Von Klitzing constant). $25,812.8074593045 \Omega$ (15 significant figures)?

We thank the Reviewer for their thorough read-through, and we have fixed these inconsistencies and errors in the main text and figures.

- 2. There are quite a few mistakes in the References. The page number of Ref. 36 is missing. The page numbers of Refs. 2, 20, and 29 must be wrong. This will almost certainly bug the Readers.

We thank the Author for spotting these errors. We have updated the references with the correct page numbers.

- 3. It is not clear to me how the Authors can compare subarray 1 (2) to a Hall bar device since the resistances of these two devices are not the same? I understand that they could tune the Q value yet could this introduce any certainties?

To compare resistor of different values we and other national metrology laboratories use a cryogenic current comparator (CCC), a device that works on the principle of superconductivity and is used in electrical metrology for highly precise comparative measurements of electric resistances. One subarray can be compared to a Hall bar, despite their different resistances, using the CCC. This is achieved by picking the appropriate current winding ratios inside the CCC, which in our case was 3776/32 as described in the methods. The coil winding error of the CCC coils can be estimated by performing an automated self-test which tests different combinations of CCC coils against each other with a 1:1 ratio comparison at high bias currents and looking for residual SQUID signal (see for instance DOI:10.1109/TIM.2020.3010111). The 64 coil is used as a reference (with an assumed zero coil winding error) in the self-test comparison sequence to measure all CCC coils (for instance $2:2$, $(2+2+4+8+16):32$, $(32+32):64$, $(64+32+32):128$, $(1024+1024):2048$ etc.) A correction for a CCC ratio error can then be estimated from the measured coil winding errors. The typical ratio error for our tests is between $0.1-0.5 \text{ n}\Omega/\Omega$ with $0.1 \text{ n}\Omega/\Omega$ uncertainty. This is well within the specifications of our CCC.

Furthermore, in the case of array vs array which use a unity CCC winding ratio 64:64, one can simply switch the subarrays so they are measured using each unique 64-winding. So in

the first comparison it would be Array1 vs Array 2 using 64_A and 64_B windings respectively, and in the 2nd comparison one can repeat Array1 vs Array2 but now connected them to 64_B and 64_A respectively. If there is a significant CCC ratio error it will be clear in the relative deviations of the subarrays. We have performed this test and saw no significant deviation outside the measurement uncertainty of $0.2 \text{ n}\Omega/\Omega$.

- 4. The Authors mentioned that "Any minor imperfection in any one individual Hall bar, be it improper quantization or poor contact resistance, will be detrimental to quantization accuracy." Could they tell the Readers how they are able to get around this issue, please?

We get around this issue by employing superconducting leads to eliminate lead resistances, split-contact design to reduce contact resistance, and use reliable fabrication methods to make the devices and dope graphene homogenously using molecular dopants. We have added additional clarifying sentences to Page 2:

"Reliable microfabrication and uniform doping is achieved using a molecular dopant technique. The influence of contact resistance and leads is reduced using multiterminal connections combined with superconducting leads"

- 5. The graphene array arrangement is the key to this nice work. However, to me, Fig. 1 (a) is not clear at all. Could they provide a schematic diagram in the supplementary information.

We have added a new Figure 1 with more detailed description of the array design.

- 6. In Fig. S1 (a), the resistance appears to increase with decreasing temperature for $T < 11 \text{ K}$. What is the reason for this?

The figure caption contains a brief explanation of this observation. The cause is due to quantum interference corrections to Drude resistance in graphene, so-called weak localization correction (see for instance our previous work DOI:10.1103/PhysRevLett.107.166602). The temperature dependence of resistance was measured for one subarray, and it is effectively two-probe configuration down on the chip, so we are measuring both the graphene channel and NbN-contact pads. Thus we see not only the superconducting transition of NbN-contacts, but also the onset of quantum corrections to resistance to graphene as lower temperatures. Note that the resistance of the subarray is around 50 Ohm since no magnetic field is on, and there is no quantum Hall effect yet.

- The base temperature of their dry TeslatronPT cryostat is 1.5 K and most of the results are obtained at $T=2 \text{ K}$. Performing the measurements at $T < 2 \text{ K}$ could help?

Lower temperatures can be better for the onset of quantum Hall effect, for instance lower magnetic field is needed and thermal noise may be reduced. However, a benefit of choosing $T = 2.1 \text{ K}$ is that we condensed liquid helium inside the VTI, and the temperature of the VTI was the most stable when He is a superfluid and near its lambda-point (only $\sim \text{mK}$ fluctuations during measurements). Furthermore, other groups (NIST) have reported that chips immersed in superfluid can have higher critical current due to better cooling (DOI:

10.1109/TIM.2018.2882958). The reference and details are briefly mentioned in the Methods section.

- 8. Finally just some format issue. $5 T^{16}$ looks like Tesla to the 16th power. They could have used 5 T (Ref. 16) instead.

This has been fixed.

Reviewer #3

Report on paper entitled

“Exceptionally accurate large graphene quantum Hall arrays for the next SI”

From He *et al*

General comment

The paper reports on the development of a Hall resistance array involving 236 single Hall channels (the single element has not the usual Hall bar geometry) made of epitaxial graphene grown on SiC by thermal decomposition and connected using NbN superconducting split-contacts, which ensure multi-terminals connection. Superconductivity of connections allows a simplification of the array design compared to the one of previously developed arrays from GaAs heterostructures. The array is divided between two subarrays of $RK/236$ ohms ($RK=h/e^2$) nominal resistance connected in series, each made of 118 Hall channels connected in parallel. The main result is the demonstration of the agreement within 0.2 parts in 10^9 of the Hall resistances measured at the terminals of the two subarrays for measurement currents as large as 5 mA. Moreover, the paper reports on the agreement of the ratio of the Hall resistances between each array and a single Hall bar with its theoretical expectation within measurement uncertainties of 2 parts in 10^9 and 0.2 parts in 10^9 as demonstrated performing indirect and direct comparisons respectively. Besides, the paper presents measurements of the ratio between the Hall resistances of the two subarrays as a function of the magnetic field around 5 T and of the current up to 10 mA.

This paper follows the two articles from Kruskopf *et al* from NIST and co-workers (Metrologia 56, 0655002 (2019), IEEE Trans. Electron devices 66,3973 (2019)), which have yet demonstrated the quality of the superconducting (NbTiN) split-contacts and the accuracy within a measurement uncertainty of 2 parts in 10^9 of a certainly more simple (only 6 Hall bars) but nevertheless large array. The main interest of this paper is the demonstration with a very low measurement uncertainty (below 10^{-9}) of the accuracy of the quantized Hall resistance of arrays made of a large number of single elements which reduces to Hall channels instead of the usual Hall bars. The design is therefore very simplified. The accuracy of the two-terminal Hall resistance of a single channel is ensured by the split-contacts, which define a local series multi-terminal connection. The accuracy of the Hall resistance of the whole array is then ensured by the superconductivity of the parallel connections (here the authors use NbN material).

This work therefore proves the reproducible fabrication of homogeneous graphene single Hall channels. Above all, it proves that the superconducting split-contacts connecting graphene do not lead to Hall discrepancies even at the highest accuracy level: detrimental effects of flux vortices or of tunnel resistance at the superconducting/graphene interface seem to be irrelevant. This is an important result for the metrological community notably in the perspective of combination of several quantum standards (Hall channels, Josephson standards, SQUID devices...). On the other hand, the simplification of the design allowed by the superconducting contacts comes at the expense of the possibility of testing quantization criteria: no measurement of the longitudinal resistance, no measurement of resistance of contacts. Contrary to the claim of the authors, the agreement of the Hall resistances of the two subarrays does not prove the accuracy of the Hall quantization. It only proves a level of reproducibility.

In this work, the proof of the quantization accuracy of the subarrays only comes from the comparison with the single Hall bar, which is itself well-characterized following the technical guidelines. Although the reproducibility of the Hall resistance varying the magnetic field and the current generally tends to confirm quantization accuracy, it does not replace measurements of the longitudinal resistance and of the contact resistance, which

are also quicker characterizations. As an example, a homogeneous current leakage (existence of a parallel finite resistance) caused by the buffer layer or the polymer deposited on top surface would lead to a quantization discrepancy not detectable by a comparison of the two subarrays. On the other hand, it would be detected by a finite increase of the longitudinal resistance (proportional to B_2). Besides, the discrepancy between the two subarrays does not allow concluding about the real deviation to quantization of each subarray. Generally, this discrepancy will be smaller than the real deviation to quantization because of partial compensation between the two subarrays. A user measuring a small discrepancy of only two parts in 10^9 between the two subarrays could not conclude about the accuracy level of the subarrays? As shown in the interesting figures S2 and S4, the two subarrays can manifest different deviations from quantization.

To conclude, this paper reports on nice metrological measurements. The results are interesting and useful for the metrological community, notably because they prove the feasibility of perfectly accurate quantum Hall simplified arrays involving a large number of elementary Hall devices connected using superconducting connections and contacts. The technology performance behind these results might interest a broader community. Nevertheless, the superconducting split-contacts in simplified arrays, but with a small numbers of elementary devices, were yet demonstrated by previous works of Kruskopf et al. Moreover, I disagree with the claim that the comparison of two subarrays is enough to prove the quantization accuracy. A user receiving the array cannot simply trust the agreement of the two subarrays Hall resistance to conclude about the accuracy level. An additional single Hall bar on the chip is in fact required to prove the quantization accuracy of the array. As proposed, this quantum device can be used as an ultra-stable secondary standard in experiments like the Kibble balance but only if periodic comparisons with a single Hall bar are carried out. In their conclusion, the authors should discuss alternative compromise design allowing measurements of the longitudinal resistance and contact resistance for quick checking, which are in principle possible for arrays based on parallel connections. Thus, although the results are of interest and confirm the perfect quantization of graphene-based arrays, I am not convinced that they deserve to be published in Nature Communications journal. A more specialized journal could be more adapted.

Authors reply to general comments:

We thank the Reviewer for this thorough discussion. We have identified the following major issues raised by the Reviewer in the preceding paragraphs (quoted again below), and provide point-by-point answers below. Many of the Reviewer's suggestion have led to changes in our main text. However, in some cases there remains a minor disagreement with the Reviewer and we hope we can convey our message below:

- *“On the other hand, the simplification of the design allowed by the superconducting contacts comes at the expense of the possibility of testing quantization criteria: no measurement of the longitudinal resistance, no measurement of resistance of contacts. Contrary to the claim of the authors, the agreement of the Hall resistances of the two subarrays does not prove the accuracy of the Hall quantization. It only proves a level of reproducibility. “*

Indeed, the agreement between the array proves a high level of quantization reproducibility. But importantly, coupled with the fact that we also demonstrate the precise quantization accuracy via comparisons to a single Hall bar (which the Reviewer recognizes), we have proven that our array is also quantized to an unprecedented degree of accuracy. The quantization can then be demonstrated by a precise resistance plateau in magnetic field sweep, since we have already checked for potential leakages via the Hall bar versus array comparison.

Furthermore, it seems that we did not clearly explain the purpose of the subarray versus subarray comparison measurement. Our intention is to propose an alternative to established quick characterization tests such as measuring R_{xx} and contact resistance of single Hall bars, which are not practically feasible for large arrays. We stress that the direct subarray versus subarray comparison is not intended to replace all other ways of characterization (it cannot). But neither can R_{xx} and contact resistance checks. We elaborate further on this point below.

- *“Although the reproducibility of the Hall resistance varying the magnetic field and the current generally tends to confirm quantization accuracy, it does not replace measurements of the longitudinal resistance and of the contact resistance, which are also quicker characterizations.”*

The R_{xx} and contact resistance tests mentioned by the Reviewer are very useful, but are by themselves also not sufficient to truly determine quantization and whether a quantum Hall device works, especially not for high-precision measurements. They are commonly used as a quick verification test for routine measurements and the paper by Delahaye (DOI:10.1088/0026-1394/40/5/302, cited in main text), which contains generally accepted guidelines, states that in order for these quick tests to work, the sample should already have been thoroughly characterized, i.e. one has to already have thoroughly demonstrated the quantization via other precision measurements. One essential test comes from comparison measurements, like those we have performed on our arrays. In the end, one must perform precision comparison measurements to find agreement between different devices, and ideally different materials and different quantum resistance values, to truly verify the quantization, which we have done. Once the quantization of the arrays has been thoroughly tested, the direct subarray versus subarray comparison measurement can then be the “quick” verification test before routine measurements. Since it utilizes CCC-measurements, it will also be more precise than R_{xx} and contact resistance checks, at the cost of increased measurement time. It is for good reason that direct comparisons between quantum Hall devices are used for the best universality tests of the quantum Hall effect, and the subarray versus subarray comparison uses the very same principles.

- *“The technology performance behind these results might interest a broader community. Nevertheless, the superconducting split-contacts in simplified arrays, but with a small numbers of elementary devices, were yet demonstrated by previous works of Kruskopf et al.”*

We have already duly acknowledged in the main text the preceding works by Kruskopf et al., along with other efforts on arrays. Additionally, we have now included even more references in the newly added Table 1. Compared to previous works, our achievements in terms of design and fabrication cannot be consider incremental for graphene arrays since the resulting devices demonstrate over an order of magnitude increase in precision coupled with over an order of magnitude more complexity in terms of number of devices and contacts. Our array device also demonstrates much higher breakdown currents, sufficient for demanding applications such as the Kibble balance which requires several milliamps.

We hope that the updated Table 1, which summarizes and benchmarks our results with respect to previous arrays, will serve to highlight the substantial increase in performance.

- *“As an example, a homogeneous current leakage (existence of a parallel finite resistance) caused by the buffer layer or the polymer deposited on top surface would lead to a quantization discrepancy not detectable by a comparison of the two subarrays. On the other*

hand, it would be detected by a finite increase of the longitudinal resistance (proportional to B^2).”

Leakages and other factors which can impact quantitation accuracy can also be detected using comparisons measurements, which are arguably more precise and sensitive methods. Future array designs can be trivially made to account for leakage tests. For instance, the subarrays could be designed to have different resistances (e.g. 10Ω and 90Ω in series for a 100Ω array), and the non-unity resistance ratio can be used to test for leakages and quantization accuracy (as suggested by guidelines by Delahaye). This can also be achieved via array versus single Hall bar comparison measurements, as was done in our work. Furthermore, these experiments, and previous studies, have yet to find significant leakage due to either buffer layer or polymer dopant. Once leakages have been tested and ruled out, the subarray versus subarray direct comparison, coupled with flatness of plateau in magnetic field sweep, is more than sufficient to verify the quantization of the array.

- “A user measuring a small discrepancy of only two parts in 10^9 between the two subarrays could not conclude about the accuracy level of the subarrays? As shown in the interesting figures S2 and S4, the two subarrays can manifest different deviations from quantization.”

See above answers on how the accuracy level of the array can in fact be concluded from subarray versus subarray measurements, provided that the necessary precautions have been taken. The figures in supplementary demonstrate that the subarray are not identical outside the quantum Hall state, with different resistances and different behavior in low magnetic fields due to different carrier density/mobility. The high-level of agreement that we demonstrate in the quantum Hall state, across a wide range of magnetic fields, can therefore only be reasonably attributed to proper quantization of the array, and not coincidental due to for instance ultra-reproducible fabrication methods. The only thing which can spoil the quantization accuracy is then a potential leakage, either in the measurement system and/or the array device, which we have disproven via the Hall bar versus array comparisons.

- “A user receiving the array cannot simply trust the agreement of the two subarrays Hall resistance to conclude about the accuracy level. An additional single Hall bar on the chip is in fact required to prove the quantization accuracy of the array. “

The above reasoning could apply to any Hall bar devices, single or arrays. For any new quantum Hall device, one should in principle perform all the rigorous tests, but once the quantization has been established, one can fall back of the faster tests such as R_{xx} , contact resistance, or indeed subarray vs subarray comparisons for routine measurements.

Throughout the long history of single quantum Hall effect devices, confidence have been established in the processes, in no small part thanks to essential direct comparison CCC-measurements between different quantized devices and quantum resistances. Graphene was not widely accepted until such tests had been done, including important ones like the universality tests with established GaAs.

See above answers on how a new array design and/or addition of a Hall bar on the chip can be used to further build confidence in arrays and allow extra verification tests if needed. Once arrays have gone through rigorous testing in the community and are established, a user can in fact trust, and verify, that the arrays are properly quantized, in the same way they can check that a single Hall bar is accurately quantized today.

- *“As proposed, this quantum device can be used as an ultra-stable secondary standard in experiments like the Kibble balance but only if periodic comparisons with a single Hall bar are carried out.”*

Here we must disagree with the reviewer. A strong point of our manuscript is to justify that the graphene quantum Hall arrays can in fact be used as primary standard. The key is that the array should be able to provide a universal, reproducible, and accurate resistance value (see e.g. DOI: 10.1088/0026-1394/41/4/010 for such discussions on why arrays can work as primary standards). We believe our experiments have demonstrated this possibility and taken crucial steps towards elevating arrays to the same level as single Hall bar standards.

We have demonstrated via rigorous comparison measurements that our array is fully quantized, and it fulfills all the criteria as a primary standard, indistinguishable from single Hall bars in terms of precision. The comparison tests with single Hall bars were used to build further confidence in the quantization accuracy. It does not make the array a secondary standard any more than the previously established GaAs Hall bars make graphene Hall bars secondary. A parallel can also be drawn to an array of Josephson junctions, which are now accepted as a primary Voltage standard alongside the single junction standard.

Periodic performance verification of an array or a single Hall bar standard are wise, especially if there is concern of sample degradation. For higher precision measurements, one should perform comparison measurements using a CCC, so checking the array against itself and/or another Hall bar is no different than checking a single graphene Hall bar standard against another standard such as GaAs, another Graphene standard, or indeed a 100 Ohm transfer standard with a known history. These comparison checks are all routine quantization tests for single Hall bar devices today, so in that respect the array does not introduce any new requirements.

Detailed comments

- 1) Figure1: The presentation of the device has to be more detailed to help the reader not expert of the multi-terminal connection scheme to understand its working: where are the two subarrays, where is the current circulating... A focus on the split-contact geometry is also required.

We have added a new Figure 1 which contain more details on the device design, and a clearer view on the split contact design. We have also added a sentence with mention of split-contact and superconducting leads, explicitly highlighting references to previous works (Page 2).

“The influence of contact resistance and leads is reduced using multiterminal connections combined with superconducting leads”

- 2) The figures of the supplementary papers are generally very interesting and should be more discussed in the main text. The figure S2, which shows the increase of deviations to quantization outside the resistance plateau, should be included in the main text. The figure S3 also gives a good summary of the results.

Supplementary Figure 2 simply demonstrates that the subarrays are not identical, with different resistances and different behavior in magnetic field due to different carrier density/mobility. This supports the notion that the high-level of agreement that we demonstrate in the quantum Hall state, across a wide range of magnetic fields, can therefore only be reasonably attributed to proper quantization of the array, and not some coincidence. The rigorous verification of this statement is then performed using hall bar versus array comparisons, which is one of the main results.

We appreciate that the Reviewer likes Supplementary Figure 3, and it does serve as a nice summary of the comparison measurements. However, we felt that the inclusion of cross-referencing different indirect and direct comparisons was ultimately not needed since we have the strongest proof of the array quantization in the precise subarray versus subarray and versus hall bar comparisons shown in the main text.

- 3) Page 2: "... without high external amplification^{6,7}". In ref 6,7, the CCC circuit is not simply used for amplification but also to decouple the external circuit involving the load from the primary circuit involving the Josephson and the Hall quantum standards only.

Our intent was to highlight the fact that using an array with lower resistance, the specific realization in the reference could be improved since the gain could be lowered, decreasing the measurement uncertainty. We have changed the sentence on Page 2 to read "without as high external amplification".

Furthermore, the array could potentially be used in combination with Josephson arrays in other ways to utilize Ohm's law to realize current, but that is outside the scope of this work. We have also added another sentence which highlights other uses of arrays in current measurements (Page 2):

"Due to their stability, QHAs are also desired for precision measurements of current in general"

- 4) Page 2: "while R_{xx} can be assessed ..., this approach is not feasible for large scale arrays". This argument is not valid for arrays based on parallel connection of Hall bars.

For strictly parallel arrays there are ways to measure the mean longitudinal resistance R_{xx} of all array elements, as was done by Poirier et al. (DOI: 10.1063/1.1495893, cited in main text). This is not possible for general array designs which contain both series and parallel connections, a necessity to achieve exact realizations of useful decade resistance values like 100Ω , $1 k\Omega$ etc. In the general case, the only way to directly determine R_{xx} is to measure it for each individual element, which is not practically feasible.

- 5) Page 2: "We propose that a direct..., is the best method... in each array element". As discussed in the general comment, I disagree with this argument, which is not proved here. The authors have to moderate the conclusions drawn from the subarray comparison.

We have clarified the intent behind the proposed method (see above). The section in the intro now reads:

"We propose that the direct comparison between two epigraphene QHAs, based on established QHE universality tests between GaAs and graphene, is the best method to verify the quantization for routine measurements, serving a similar purpose as measuring

R_{xx} and checking contact resistance in individual Hall bars. In the end, our measurements show that large arrays can meet the stringent criteria set by single Hall bar metrological standards, and QHAs can be used as a primary standard which can exceed traditional single Hall bar devices in terms of applications.”

We stress that the main conclusion should be that we have proven that our record-size graphene array is quantized with unprecedented precision. In addition to that, we demonstrate that the subarray versus subarray comparison can be used to test the quantization with high precision, and we demonstrate that there is no “hidden” influence on the quantization accuracy of arrays by performing single Hall bar versus array comparisons.

- 6) Page 3: *“This degree of accuracy in the quantization of such a large QHA...”*
*NO, this is a measurement of the reproducibility. This is not a proof of accuracy. The accuracy is proved by the comparison of the array with the single Hall bar. Here, add Ref 3, which also reports on a graphene/GaAs universality test with a record uncertainty: 8.2x10⁻¹¹. Concerning test of the reproducibility of the Hall resistance in single Hall bars, paper “F. Schopfer et al, JAP, **114**, 064508 (2013)” reports on the lowest uncertainty (3x10⁻¹¹).*

We thank the reviewer for this important distinction. We have demonstrated both quantization reproducibility via subarray versus subarray comparisons, and quantization accuracy via Hall bar versus subarray, without any detectable discrepancies, to a high precision of 0.2 nΩ/Ω. We can therefore confidently claim such a degree of accuracy for our arrays. However, we have taken care to change “accurate” to “precise/reproducible” when talking solely about array versus array comparisons. We have also explicitly mentioned “quantization accuracy” in the hall bar comparison section.

The suggested reference by Schopfer contains very nice Wheatstone bridge measurements, but the article has not demonstrated an Allan deviation which experimentally supports such a low uncertainty. The stated ultra-low uncertainties are derived from extrapolations to zero dissipation and should be taken as an ideal case, also assuming that there is only white noise. The best experimental value of the white noise limit for quantum Hall measurements is perhaps reported by R. Ribeiro-Palau et. al (DOI:10.1038/nnano.2015.192, cited in main text) where they show that white noise dominates down to almost 0.1 nOhm/Ohm. This is why we only claim our experimentally motivated uncertainty of 0.2 nOhm/Ohm in this work. However, we have included a related reference by Schopfer (DOI: 10.1063/1.2776371), which the Reviewer’s suggested reference is based on, since it deals with universality tests using Wheatstone bridge approach. There they show that the Allan deviation becomes dominated by uncertainty below a level of 0.1 nOhm/Ohm. We have added the following sentence to accompany the reference (Page 5):

“Another type of universality test was performed using a small 4 Hall bar GaAs array in a Whetstone bridge setup, which tested the reproducibility of the QHE with an uncertainty down to even 0.076 nΩ/Ω. Note however that in all cases the experimentally motivated level of uncertainty from Allan deviation is at best around 0.1 nΩ/Ω, comparable to our value of 0.2 nΩ/Ω”

- 7) page 6. *“The quantization was tested by performing precision measurements at different fields”. Here, “quantization” has to be replaced by “reproducibility” because these results report on a comparison of the two subarrays.*

We have corrected this, in line with the discussion above.

Reviewer #4 (Remarks to the Author):

The authors describe the implementation of a new large-scale quantum Hall array resistance standard based on graphene on SiC and superconductive contacts. Experimental data display a reproducibility of 0.2 ppb and a large drive current. One possible application is the redefinition of the kilogram unit using a Kibble balance with an integrated resistance standard.

The manuscript is very interesting and timely and, to me, it looks suitable for publication. I just report down here a few minor points of improvement:

- 1) The authors mention that carrier density in their device is obtained by molecular doping, and measurements indicate $1.7 \times 10^{11} \text{ cm}^{-2}$ carrier density (a bit hidden in the caption to Fig.2). I could not find any indication on the expected homogeneity of the doping, Methods section just states the doping is "homogeneous". Any estimate of the homogeneity? Is it a relevant improvement parameter here? Given the accuracy, I guess all the graphene bars are well into the quantum regime as intended.

One way to quantify the homogeneity of this doping on graphene is by measuring how close one can approach the Dirac point in the material (zero charge carrier scenario). A perfectly homogeneous doping will result in (hypothetically) being able to reach zero carriers in graphene, whereas inhomogeneous doping would result in high residual doping when trying to reach the Dirac point.

In previous works we have demonstrated that the chemical doping method results in a residual doping below 10^{10} cm^{-2} , see He et. al (DOI:10.1038/s41467-018-06352-5, cited in main text). In this work, we did not explicitly describe homogeneity further since it would be clear from precision measurements whether all array elements were quantized or not. Any significant inhomogeneity would spoil the quantum Hall effect and impede the high-precision measurements we demonstrate in this work. This small inhomogeneity is manifested in the slightly different behavior of the subarrays in non-quantizing state, as demonstrated in Supplementary section.

We added explicit mention of the homogeneity in the Methods section:

"This ensures a stable, homogenous, and controllable doping over the whole chip, with an expected charge inhomogeneity of doped graphene being below 10^{10} cm^{-2} ."

- 2) The device studied by the authors is interesting and somewhat non-conventional with respect to the ones typically used for QH standards. I believe that the term "Hall bar" might look a bit misleading to some readers: for instance, in a Hall bar, I would naively expect lateral contacts with independent wiring that allow proper 4 wire measurements, while here the device is rather built by putting many graphene conductors (each one with two "split contacts") connected in parallel. The four-wire measurement scheme is then realized by taking advantage of the zero-resistance superconductive leads... but basically this is a set of two-wire graphene conductors in parallel. So for instance R_{xx} cannot be measured not much because the array is large: given this contacting geometry (without independent lateral

contacts) R_{xx} would be impossible to measure even if the array contained just a single graphene element!

With the inclusion of Figure 1 this will hopefully aid future readers in their understanding of our array geometry. We use the traditional name of “Hall bar” to describe one array element, which is topologically equivalent to a standard Hall bar and for all intents and purposes functions the same. We are indeed effectively measuring two-probe resistance, which in the quantum Hall state (with zero contact resistance) is $h/2e^2$.

On the note of R_{xx} , one could in principle have added additional individual current and voltage pads for each parallel Hall bar/element, send current only through each one at a time, and measure the R_{xx} of all of them separately. Due to the amount of individual elements, this is of course not feasible in practice. This is why we only have 2 split contacts per array Hall bar/element, because there is no need to probe the R_{xx} of each individual Hall element. We instead perform comparison measurement to validate the performance of our array. See also our reply to Reviewer #3 for more discussions on this topic.

- I don't want to say this naming is "wrong": one can always think about the lateral contacts as shorted to the source and drain contacts of the Hall bar, or maybe just a "split-contact"; however, the authors could probably spend few words to communicate how this is not really a conventional Hall bar. I would anyway stress that the manuscript contains recent references 19 and 20 about these details, but that's a bit cryptic.

The Reviewer is correct in thinking that lateral contacts can be thought of being shorted to source/drain, forming a split-contact. Traditionally, a Hall bar is a four-terminal (or more) device, but we feel that this term is also applicable for the elements of the arrays, even if they are effectively two-probe devices. Each element is equivalent to a Hall bar wired in the multiple-connection configuration, and as the Reviewer points out, the details are described in the references in the main text (e.g. DOI: 10.1063/1.353944). We have added above clarification in the caption of the new Figure 1.

- I also wonder if - beyond eliminating the lead resistance - the use of superconductive electrodes also contributes to minimizing the contact resistance to graphene of the split-drain and split-source. But this discussion would probably go out of the scope of the paper.

It is entirely possible that this is the case, but the details are left to future work. We are currently working on a separate publication related to contacts to graphene, which might be of interest in the future.

- 3) The paper contains some typos and broken sentences. I suggest a careful re-read. Few examples:

- caption to Fig.1. "... Allan deviation no longer decreases due to $1/f$ noise and drift (starts?) dominating. The error bars are estimated relative errors (?)."

- page 2 line 4, "they could using Ohm's law be combined...", maybe just a shuffling here.

We thank the reviewer for spotting these errors. We have corrected them, and others. For

instance, the sentence in Fig 2 (previously Fig 1) and 4 now reads “The error bars are estimated relative errors (see Methods)”.

REVIEWER COMMENTS

Reviewer #1 (Remarks to the Author):

I would like to thank the authors for modifications that improve the transparency of the present work. The results are impressive and further the effort to make quantum resistance standards more easily available and more useful. I find that the inclusion of past efforts presented in the new table are somewhat limited, because they only show some of the array work, but this may be an oversight.

The authors do not cite useful work aimed at reducing the number of units in the array by optimizing the design to conserve "real estate" on the chip, which has been of interest to other researchers. These efforts have addressed the interest in producing arrays of decade value, which can be important in existing scaling methods.

My decision for publication is that this manuscript would be better suited to a more specialized journal that focuses on the particular problems of metrology. This is due to the lack of new processes or methods that contribute to improving the technology of quantum standards, and not the data, which is very impressive in its support of the technology. Detailed comments follow.

Second review (04/25/2022)

P3: The statement, " we propose that the direct comparison between two epigraphene QHAs, based on established QHE universality tests between GaAs and graphene^{3,32,33}, is the best method to verify the quantization for routine measurements, serving a similar purpose as measuring RXX and checking contact resistance in individual Hall bars" is misleading, as noted by another reviewer. Contrast this with the following statement from a 2020 CPEM summary paper, "We introduce a dc QHARS design that uses the Wheatstone bridge principle as a precise test of the QHARS self-consistency for dc measurements." Self-consistency is not a proof of universality, at least not at the highest level of metrology.

More specifically, why is a single comparison between two equal-value subarrays claimed as the best? Using the CCC method of resistance ratios, the one-to-one ratio limits the measurement uncertainty compared to larger ratios where the SQUID feedback is applied to a small number of windings. The quantization of two similar arrays on a single chip also may deviate from the ideal value in tandem, such that nearly equal deviations occur in both sub-arrays, and equality of the values is observed.

P3: The reference 34 is given to describe the equilibration length of hot electrons. This reference develops a model for cooling of hot electrons in a chiral edge state, specifically "for a mechanically free edge of graphene in devices where a flake is bound to the substrate by van der Waals forces" and may not apply to epitaxial graphene, where cooling mechanisms via the buffer layer are present.

Reviewer #2 (Remarks to the Author):

The Authors have adequately addressed all the issues raised by me. As a result, the revised manuscript reads much better. Therefore, I can now recommend the publication of this paper in Nature Communications. I trust that this paper will make an impact in the research fields of graphene and the new SI.

Report on revised paper (NCOMMS-21-51410A) entitled

“Exceptionally accurate large graphene quantum Hall arrays for the next SI”

From He *et al*

General comment

The paper has been noticeably improved after consideration of some remarks of the reviewers. **However, the paper as written still claims that the agreement of the resistance values of the two subarrays remains the main demonstration of the “array accuracy” in this work, which is not true!**

The sentence page 6-“We have also compared the subarrays to an on-chip single Hall bar, in order **to further verify** the quantization accuracy” as well as the paragraph order (first subarrays comparison, second subarray/Hall bar comparison) lets the reader think that the subarrays comparison could have been sufficient.

The author response “*We have demonstrated via rigorous comparison measurements that our array is fully quantized, and it fulfills all the criteria as a primary standard, indistinguishable from single Hall bars in terms of precision. **The comparison tests with single Hall bars were used to build further confidence in the quantization accuracy.***” confirms this claim.

Once again, **a primary quantum standard is a standard, which must be autonomous**: before use no extra comparison with another quantum standard is required to set that it is perfectly quantized. For example, although some single-electron tunneling devices were demonstrated to deliver quantized currents within a few 10^{-7} , these quantum sources are not considered to be primary quantum standards of current, at the present time. The reason is that it doesn't exist quantization criteria, the checking of which would allow the determination of the level of current quantization for a particular device!

In this paper, the authors have very well demonstrated the Hall resistance quantization of a particular array through the comparison with a single Hall bar, the quantization of which was itself established by the checking of the quantization criteria following the technical guidelines. Then, the subarrays comparison with variation of magnetic field and current builds further confidence in the quantization accuracy. From these results, the authors can finally make the proposal that subarrays comparison could replace the usual quantization criteria “ R_{xx} and contacts measurements”. **But, this should appear as a proposal and not an evidence! To my opinion, the user of the array must be able to perform a comparison with a reference Hall bar (one that can be included on the chip of the array) to confirm the quantization accuracy of the array.** Concerning this point, I would say that large previous QHAs were more trustable because R_{xx} and contacts measurements were possible: **Thus, I disagree with the form of the sentence** “page 2-Large QHAs have not until now truly met the stringent criteria set single Hall bar primary metrological standards in terms of precision, reliability, and reproducibility”. Besides, this paper does not demonstrate the reproducibility of the device fabrication of proposed arrays.

Reviewer #4 (Remarks to the Author):

The authors have solved the few doubts I had about the work. In particular, the homogeneity of the 2D electron system has now been clarified. I am positive towards the publication of the manuscript.

Main remark/comment:

In order to avoid any confusion about this crucial point, the authors should organize their paper differently: 1) demonstration of the array quantization by comparison with a Hall bar, 2) report on the subarray comparisons, 3) make the proposal that the subarray comparison could be used as a quantization criterion. In the table 1, the main result should be the comparison of the array with a Hall bar $(-0.04 \pm 0.2) \text{ n}\Omega / \Omega$. 4) The sentence on page 2 “Large QHAs have not until now truly met the stringent criteria set single Hall bar primary metrological standards in terms of precision, reliability, and reproducibility” should be moderated.

Conclusion:

This work demonstrates with a record uncertainty (for an array) the quantization of the resistance of an array made of a large numbers of Hall bars with a simplified design. On one hand, as commented by referee 1, this work builds on previous demonstrations by NIST that contacts between graphene and superconductors can work under magnetic field. On the other hand, it proves that these superconducting contacts allow the array achieving the best accuracy.

The authors propose that the comparison of resistance of subarrays (with variation of magnetic field and current) could replace the usual quantization criteria. To my opinion, this is not obvious and requires further demonstrations but it is the merit of authors to make this proposal. On this subject, I regret that the authors have not discussed alternative and improved designs allowing the checking of contacts/ R_{xx} . Nevertheless, even if a comparison with a Hall bar should be required before any use, this kind of array can be very useful for applications. **In the end, I keep some reservations about publishing this work in the Nature Communications journal and in any case, I cannot recommend publication before consideration of my main remark.**

Detailed remarks

1) Kibble balance compares powers and not forces. Thus, the authors should replace “determines the gravitational force acting on an objectusing the measured local gravitational acceleration”

by something like “compares the mechanical power of a mass moving at a velocity v under the gravitational acceleration g in terms of the electrical power of a measured current I under a voltage V .”

2) Page 5. Another type of universality test..... with an uncertainty down to even $0.076 \text{ n}\Omega / \Omega$. The best reproducibility test of the QHE, supported by Allan deviation analysis, was

achieved with an uncertainty down to $0.031 \text{ n}\Omega/\Omega$. (F. Schopfer et al, J. Appl. Phys, 114, 064508, 2013).

3) Page 8. "The deviation at 8.5 mA is within $1 \text{ n}\Omega/\Omega$ at lower magnetic fields $< 5\text{T}$, which is acceptable for most practical metrological applications, including the Kibble balance".

The relative deviation between both array values is within $1 \text{ n}\Omega/\Omega$ but the discrepancy between each array value to the nominal value is larger as shown by figure S4 (rather $3 \text{ n}\Omega/\Omega$). Again, this is this latter discrepancy which is important for applications! The sentence must be corrected.

4) Many quantities must be italicized: fundamental constants, R_K, \dots

5) When expressing quantities in tesla unit, replace magnetic field by magnetic induction!

Point-by-point reply

We would like to thank the two Reviewers for their additional questions. The Reviewers' comments are reproduced verbatim below and are written using normal font. The authors' answers are given using both bold and italic font. We have also formatted each question and answer into its own bullet point for better readability. The changes in the revised manuscript are marked with yellow highlights.

Reviewer #1 (Remarks to the Author):

I would like to thank the authors for modifications that improve the transparency of the present work. The results are impressive and further the effort to make quantum resistance standards more easily available and more useful. I find that the inclusion of past efforts presented in the new table are somewhat limited, because they only show some of the array work, but this may be an oversight.

The authors do not cite useful work aimed at reducing the number of units in the array by optimizing the design to conserve "real estate" on the chip, which has been of interest to other researchers. These efforts have addressed the interest in producing arrays of decade value, which can be important in existing scaling methods.

The cited works (e.g. summarized in Table 1) cover a wide range of arrays design with different elements and resistances. Where possible, we have limited our choice to those works which have the best precision measurements in order to have a fair comparison to our work. This might give the impression some other works have been omitted. We welcome the Reviewer to bring any interesting and relevant works to light and we will be happy to consider adding them to Table 1.

Our work represents the latest developments in array technology, which is primarily focused on large arrays with many elements. Due to their complexity, large and precise arrays have been difficult to realize in practice until now. Large arrays are integral since they are needed to realize the widest range of useful resistances values, especially lower resistance values like 100 Ohm or below, which are crucial for metrology. Fabricating and testing large arrays present the highest challenge, and once a method has been proven, it can be easily adapted to smaller arrays if chip-size is limited.

My decision for publication is that this manuscript would be better suited to a more specialized journal that focuses on the particular problems of metrology. This is due to the lack of new processes or methods that contribute to improving the technology of quantum standards, and not the data, which is very impressive in its support of the technology. Detailed comments follow.

As the Reviewer acknowledges, our data support the fact that our graphene arrays are impressive and further the effort to make quantum resistance standards more easily available and more useful.

We have placed the focus on the subarray comparison measurements, which are the first time such precise measurements have been performed on such large arrays. An obvious new contribution is our suggestion to add to the existing guidelines for quantum resistance metrology, which need to be updated to meet the new demand of arrays.

These arrays would not be possible without the development of new techniques to complement previous methods. The Reviewer's comments have stimulated us to highlight the importance of the processes and methods used to fabricate such arrays, and this has resulted in us writing a new manuscript based solely on the new processes and methods in our work. This manuscript is currently submitted to a more specialized journal and available as pre-print (<https://arxiv.org/abs/2206.03839>). We have also added a citation to our pre-print in the main text:

"The NbN was deposited using sputtering, and was used in combination with a special fabrication method to create superconducting edge contacts to graphene (<https://arxiv.org/abs/2206.03839>, see Methods)"

In a nutshell, our way to produce electrical contacts to graphene is based on our novel scalable way to fabricate edge contacts to 2D-materials. In our pre-print we describe contacts to graphene with normal metals achieved by metal evaporation, while in this manuscript we show that our process/method also allows superconducting materials to be deposited by sputtering. The large array, with around 500 contacts, represents the ultimate test for this process/method, and we have demonstrated in this work that the fabrication quality is indeed flawless.

In conclusion, we maintain the claim that our work certainly offers novel process and methods which radically improve the technology of quantum standards.

Second review (04/25/2022)

P3: The statement, "we propose that the direct comparison between two epigraphene QHAs, based on established QHE universality tests between GaAs and graphene^{3,32,33}, is the best method to verify the quantization for routine measurements, serving a similar purpose as measuring RXX and checking contact resistance in individual Hall bars" is misleading, as noted by another reviewer. Contrast this with the following statement from a 2020 CPEM summary paper, "We introduce a dc QHARS design that uses the Wheatstone bridge principle as a precise test of the QHARS self-consistency for dc measurements." Self-consistency is not a proof of universality, at least not at the highest level of metrology.

The above simply states that we feel that the subarray comparison is the best, i.e. most precise/trustworthy, method to verify that the array is still in working condition for routine measurements. As we have discussed in-depth with Reviewer 3 in the first round of replies, we are primarily aiming to provide new methods for graphene arrays in metrology, which are currently not included in the existing guidelines for quantum resistance standards. We are not claiming to or aiming to replace existing methods, and once should still take all necessary precaution to check the system in its entirety, including stronger universality tests via comparison to other devices and quantum resistances. See previous replies to Reviewer #3 for more details.

To avoid further confusion, we have altered the statement slightly:

“...it is the most precise and reliable method to test the quantization for routine measurements”

Regarding universality test, the two subarrays have been shown to be physically different (carrier density etc.) and do serve as a universality test of the quantum Hall effect. Of course, it is not as strong as comparing GaAs to graphene for instance. We are therefore not making any claims on universality testing in this work, only that the comparison method is based on the same principle as universality tests.

More specifically, why is a single comparison between two equal-value subarrays claimed as the best? Using the CCC method of resistance ratios, the one-to-one ratio limits the measurement uncertainty compared to larger ratios where the SQUID feedback is applied to a small number of windings. The quantization of two similar arrays on a single chip also may deviate from the ideal value in tandem, such that nearly equal deviations occur in both subarrays, and equality of the values is observed.

It is precisely due to the low uncertainty of direct one-to-one ratio comparisons that we chose to make the subarrays identical in resistance. We can then determine their agreement and the quantization reproducibility to the highest level of precision. We have since replaced “best” with “precise and reliable”.

As discussed with Reviewer #3 in the previous round of replies, one can check for quantization accuracy via comparison to a single Hall bar for instance. Alternatively, for future arrays one could also simply make the subarrays different in resistance. Then the subarray comparison could also test for quantization accuracy and detect potential leakages for instance.

The argument that agreement from comparison measurements can be due to pure coincidence can be used against all measurements, since one can never be 100 % sure due to finite uncertainties. However, the notion that subarrays would deviate from their ideal value in tandem is extremely improbable, since the agreement we observe is on the order of 10 nOhm, consistent across different bias currents and magnetic fields. Additionally, we demonstrate in the Supplementary information that the carrier densities of the subarrays are different, and it is known from previous works that the charge disorder from the chemical doping technique is on the order of 10^{10} carriers per cm^2 . This corresponds to several percent deviation in normal resistance (several ohms) at our carrier densities (See: DOI: 10.1038/s41467-018-06352-5). The sub-part-per-billion agreement that we see, for various measurement conditions, can therefore only reasonably come from both subarrays being fully in the quantum Hall state. Furthermore, our subarray versus single Hall bar comparison ruled out additional leakages and errors, so we can also claim the quantization is accurate to a similar level. Future arrays designs can also be made with subarrays with different resistances, which makes the comparison measurement even more sensitive to potential errors, since they would affect the subarrays differently (different number of elements and/or different current through individual elements). We have also addressed this point in our previous reply to Reviewer #3, and that reply can be referenced for more

details.

P3: The reference 34 is given to describe the equilibration length of hot electrons. This reference develops a model for cooling of hot electrons in a chiral edge state, specifically “for a mechanically free edge of graphene in devices where a flake is bound to the substrate by van der Waals forces” and may not apply to epitaxial graphene, where cooling mechanisms via the buffer layer are present.

The Reviewer is correct that our graphene system is a bit different. However, from our internal experiments, by varying the distance between source contact and voltage probes, the equilibration length is on the order 80 nm, which corresponds well to the theory.

Reviewer #3 (Remarks to the Author):

General comments

The paper has been noticeably improved after consideration of some remarks of the reviewers. However, the paper as written still claims that the agreement of the resistance values of the two subarrays remains the main demonstration of the “array accuracy” in this work, which is not true!

We believe that the confusion with accuracy/reproducibility has been fixed after our previous discussion with the Reviewer. We reiterate that argument of “array accuracy” is derived from the comparison of subarray vs single Hall bar. Recall that the hall bar versus array test yielded a comparable degree of precision as array versus array, which is perhaps where the confusion comes from. We have now further clarified the importance of array versus hall bar test in the introductory sections and in the modified summary/conclusion.

“In summary, we have demonstrated that a record-sized graphene QHA (236 Hall bars) is quantized with an unprecedented precision of 0.2 nOhm/Ohm verified by traditional comparisons to a single Hall bar device...”

The sentence page 6-“We have also compared the subarrays to an on-chip single Hall bar, in order to further verify the quantization accuracy” as well as the paragraph order (first subarrays comparison, second subarray/Hall bar comparison) lets the reader think that the subarrays comparison could have been sufficient.

The author response “We have demonstrated via rigorous comparison measurements that our array is fully quantized, and it fulfills all the criteria as a primary standard, indistinguishable from single Hall bars in terms of precision. The comparison tests with single Hall bars were used to build further confidence in the quantization accuracy.” confirms this claim.

We have removed the word “further” to make it clearer. The sentence now reads: “We have also compared the subarrays to an on-chip single Hall bar, in order to verify the quantization accuracy”

Once again, a primary quantum standard is a standard, which must be autonomous: before use no extra comparison with another quantum standard is required to set that it is perfectly quantized. For example, although some single-electron tunneling devices were demonstrated to deliver quantized currents within a few 10^{-7} , these quantum sources are not considered to be primary quantum standards of current, at the present time. The reason is that it doesn't exist quantization criteria, the checking of which would allow the determination of the level of current quantization for a particular device!

The example with single-electron tunneling devices is not applicable here, since the reason they are not considered primary standards is that the accuracy of the output current needs to be verified with a ammeter, which is traceable to resistance and voltage (and not current!).

In our case, the comparison measurement we are performing is not a calibration of the array resistance in that sense. We are comparing two quantum Hall standards, in accordance to established guidelines (Delahaye, DOI: 10.1088/0026-1394/40/5/302). This direct quote from the guidelines explicitly states that comparison measurements are essential for primary standards: “A last but essential criterion for judging a particular quantum Hall resistance measurement is its agreement with measurements made on other devices, preferably from different wafers, and with different quantum numbers”.

A practical quantum standard is more than the perfect quantum theory it is based on, but one must also consider the material which manifests the phenomena, the electrical leads, the measurement electronics, and more. All these external factors can influence the quantization accuracy (e.g. potential leakage paths, as Reviewer #3 alluded to last time) and it is therefore important to test the whole system when using primary standards. Indeed, for practical metrology primary standards are compared to other primary standards all the time. Before a new primary single Hall bar standard can be used for the first time, one always compares it to previous established standards with a known history (e.g. GaAs versus Graphene). Furthermore, published results which report high-precision measurements on quantum Hall standards always use comparison measurements, and with good reason. We hope that it is clear now why we chose to focus on comparison measurements.

The quantization criteria of arrays have not been established yet, which is one of the motivations behind our work here. The existing guidelines were made with single Hall bars in mind, and they are not axioms. They are methods developed and agreed upon by the community to ensure reliable and reproducible realizations. They are defined based on the idea that quantization is closely related to universality. If a quantum standard is independent of the measurement parameters (temperature, magnetic field, current, device geometry, material etc.), it is expected that its value is determined by only universal constants. As we have demonstrated, this is entirely possible to achieve for arrays, but the community must now develop these criteria together (See discussion on this topic for arrays DOI: 10.1088/0026-1394/41/4/010).

For arrays, some of the methods established in the old guidelines can be transferred and adapted. One can still check the reproducibility of quantization (flatness) in magnetic field, current, and temperature. One can also perform comparison measurements with other quantum resistances (other arrays/subarrays) to check for accuracy. We have performed these tests in our work. The only things which are not suitable for arrays are the R_{xx} and contact resistance measurements, but we argue that they are not necessary and subarray comparisons can take their place as a quick test for routine measurements, provided that the other complementary measurements have been performed. See below for more details on this point.

In this paper, the authors have very well demonstrated the Hall resistance quantization of a particular array through the comparison with a single Hall bar, the quantization of which was itself established by the checking of the quantization criteria following the technical guidelines. Then, the subarrays comparison with variation of magnetic field and current builds further confidence in the quantization accuracy. From these results, the authors can

finally make the proposal that subarrays comparison could replace the usual quantization criteria “Rxx and contacts measurements”. But, this should appear as a proposal and not an evidence!

Perhaps this is a discussion on style of presenting the results. Since we believe one of the most interesting result of our work is the precise array vs array comparison (first of its kind, in terms of scale and precision), we have chosen to present it first. The precise quantization of the array is then supported by the comparison of array vs Hall bar (accuracy check). The Reviewer in fact acknowledges that this is perfectly valid.

Using the array versus array/Hall bar comparison measurements as evidence, we do propose that subarray comparison can take the place of Rxx and contact resistance for routine measurements of arrays. In the main text, we have exclusively used words such as “suggest” and “propose” for this claim. However, we have added further mention that this is a suggestion in the new summary/conclusion.

“...The proposed method of direct comparison of subarrays could be considered as an addition in future versions of practical metrological guidelines, which are in need of revision given the new wave of quantum metrology devices based on epigraphene...”

To my opinion, the user of the array must be able to perform a comparison with a reference Hall bar (one that can be included on the chip of the array) to confirm the quantization accuracy of the array. Concerning this point, I would say that large previous QHAs were more trustable because Rxx and contacts measurements were possible: Thus, I disagree with the form of the sentence “page 2-Large QHAs have not until now truly met the stringent criteria set single Hall bar primary metrological standards in terms of precision, reliability, and reproducibility”.

It is trivial to include a single-Hall bar on the chip, to maintain compatibility with the current guidelines. However, we would like to argue that once the use of arrays is more accepted by the community, and with new guidelines, we foresee that the single hall bar device could be omitted. One could instead have an array consisting of two subarrays with different resistances, since a comparison measurement between those would also check the quantization accuracy. As we mentioned last time, there are also ways to measure the mean Rxx of the whole array (DOI: 10.1088/0026-1394/41/4/010).

While Rxx and contact resistance measurements are useful for single Hall bars, they are not more trustworthy than comparison measurements, and come with greater uncertainties, especially if they were to be used for arrays.

The reason Rxx can be useful is that there is a connection between Rxx and the quantized Hall resistance. If Rxx is measured to be large, one can expect a certain deviation of Rxy from its nominal quantized value. However, this coupling is far from universal, and depends on temperature, magnetic field, geometry, charge homogeneity and more (DOI: 10.1088/0026-1394/40/5/302). Even in the best cases, Rxx cannot be used to determine the quantization accuracy below parts-per-billion for single Hall bars (see DOI: 10.1063/1.2776371 and DOI: 10.1038/nnano.2015.192). Furthermore, the guidelines explicitly state that “...comparisons test for shunting resistances (parallel

conduction) across the device, especially those between source and drain that are not revealed by Rxx measurements.”. These are some of the potential leakages brought up by Reviewer #3 in the past, which cannot be detected using Rxx-measurements. Deviation from quantization accuracy can be determined, with greater accuracy and while checking for leakages, via comparison measurements.

The contact resistance is measured according to guidelines using a 3-probe measurement in quantizing conditions. The largest uncertainty contribution comes from the fact that the wire and lead resistance need to be subtracted to get the contact resistance, and the results are therefore usually not very precise (100 mOhms uncertainty are typically reported). For a single Hall bar, the measurement is not too difficult since an acceptable contact resistance is on the order of 10 ohm. However, our arrays are much more sensitive, and even if a single Hall bar out of the 118 in parallel has a contact resistance of just 1.5 mOhm (with the rest of the contacts, including leads, being 0 resistance!) this will result in an appreciable deviation from nominal value by 1 nOhm/Ohm! Since such small contact resistances are hardly possible to accurately determine using the old method (also too many contacts to measure), and one can instead directly check the quantization accuracy via comparison measurements.

Besides, this paper does not demonstrate the reproducibility of the device fabrication of proposed arrays.

The high precision in our measurements of quantization already demonstrate essentially perfect device yield for 236 individual Hall bars with extremely low contact resistance, and low residual mean Rxx. This is already a proof of reproducibility in device fabrication. Keep in mind that due to the two-probe nature of array elements, any unwanted resistances from leads/contacts enter directly into the quantized resistance of the array.

Based on this comment, and comments from Reviewer #1, we wish to further highlight the importance of our process/method to fabricate such arrays, and this stimulated us to write a new manuscript based solely on the new process/methods in our work. In brief, our unique way to produce electrical contacts to graphene is based on our novel scalable way to fabricate edge contacts to 2D-materials, and this has spawned an entirely new manuscript, under review elsewhere and available as pre-print (<https://arxiv.org/abs/2206.03839>, cited in main text).

Furthermore, we have made 3 other smaller array devices prior to this work (max 10 array elements), and they all worked just as well. Finally, since completing this work, we have also fabricated and tested a new, and even larger array, with nominal value of exactly 100 Ohm (over 300 individual Hall bars). Precision comparison measurements between those subarrays shows excellent quantization, and this work will be published later this year.

Our work represents the first important steps for large precise arrays and is understandably limited in the number of tested devices. We invite all members of the community to reproduce our results, and test similar arrays using their setup, with the aim to perform inter-laboratory comparison measurements in the end. The goal is to build confidence in these graphene arrays, so that they will be accepted and established just like their single Hall bar counterparts.

Main remark/comment:

In order to avoid any confusion about this crucial point, the authors should organize their paper differently: 1) demonstration of the array quantization by comparison with a Hall bar, 2) report on the subarray comparisons, 3) make the proposal that the subarray comparison could be used as a quantization criterion. In the table 1, the main result should be the comparison of the array with a Hall bar (-0.04+/-0.2) nOhm /Ohm. 4) The sentence on page 2 "Large QHAs have not until now truly met the stringent criteria set single Hall bar primary metrological standards in terms of precision, reliability, and reproducibility" should be moderated.

We appreciate the Reviewer's suggestions, but we feel that the logical structure of our paper is sound. We wish to emphasize the large subarray versus subarray precision comparison measurement, since it is the first of its kind, and basis for suggested addition to guidelines on arrays. Since three other reviewers agree with the logical structure of our manuscript and share our sentiment, we choose to refrain from making sweeping changes. We hope this is understandable.

Following the Reviewer's suggestion, we have tempered the claims and altered the sentence to:

"However, large QHAs have not until now matched the performance achieved by single Hall bar primary metrological standards in terms of precision, reliability, and reproducibility"

Conclusion:

This work demonstrates with a record uncertainty (for an array) the quantization of the resistance of an array made of a large numbers of Hall bars with a simplified design. On one hand, as commented by referee 1, this work builds on previous demonstrations by NIST that contacts between graphene and superconductors can work under magnetic field. On the other hand, it proves that these superconducting contacts allow the array achieving the best accuracy.

The authors propose that the comparison of resistance of subarrays (with variation of magnetic field and current) could replace the usual quantization criteria. To my opinion, this is not obvious and requires further demonstrations but it is the merit of authors to make this proposal. On this subject, I regret that the authors have not discussed alternative and improved designs allowing the checking of contacts/ R_{xx} . Nevertheless, even if a comparison with a Hall bar should be required before any use, this kind of array can be very useful for applications. In the end, I keep some reservations about publishing this work in the Nature Communications journal and in any case, I cannot recommend publication before consideration of my main remark.

We stress again that we are not aiming to replace existing criteria, but to provide a better alternative which is more useful for arrays. Our suggestions are completely compatible with existing guidelines. As we discuss above, we believe that the subarray comparisons can fulfill a similar role to R_{xx} and contact resistance to test quantization for routine measurements, and is in many ways more precise.

Detailed remarks

Kibble balance compares powers and not forces. Thus, the authors should replace “determines the gravitational force acting on an objectusing the measured local gravitational acceleration” by something like “compares the mechanical power of a mass moving at a velocity v under the gravitational acceleration g in terms of the electrical power of a measured current I under a voltage V .”

Following the Reviewer’s suggestion, we have altered the statement to the following:

“...compares the mechanical power of a mass, moving at a certain velocity under the gravitational acceleration, to an electrical power determined by measured current I and voltage V ”

2) Page 5. Another type of universality test..... with an uncertainty down to even 0.076 nOhm/Ohm The best reproducibility test of the QHE, supported by Allan deviation analysis, was achieved with an uncertainty down to 0.031 nOhm/Ohm (F. Schopfer et al, J. Appl. Phys, 114, 064508, 2013).

The paper the Reviewer suggests contains no Allan deviation data. The only mention of Allan deviation is the sentence: “Allan variance analysis shows that the standard mean deviation is an adequate estimate of the uncertainty (noise is white, no 1/f noise is observable)”. It does not explicitly state what level of uncertainty was observed, just that it was “adequate”. Since there is no experimental data, we stand by our statement. See also our mention of this paper in our first round of replies.

3) Page 8. “The deviation at 8.5 mA is within 1 nOhm/Ohm at lower magnetic fields < 5T, which is acceptable for most practical metrological applications, including the Kibble balance”. The relative deviation between both array values is within 1 nOhm/Ohm but the discrepancy between each array value to the nominal value is larger as shown by figure S4 (rather 3 nOhm/Ohm. Again, this is this latter discrepancy which is important for applications! The sentence must be corrected.

Figure S4 shows comparison measurements against a secondary standard with 12.9 kOhm resistance (we have since clarified this further in the figure caption). Since it is not a quantum standard, the measurements have more uncertainty/noise. We are comparing results taken at 3 mA, 5 mA and 8.5 mA, which show no discrepancies within the uncertainty of 3 nOhm/Ohm ($k=2$). Since we know that 3 mA and 5 mA are well-quantized, the quantization at 8.5 mA also appear to be good, limited by noise. Much of the uncertainty comes from the 3 mA measurement. If we compare only 5 mA to 8.5 mA, the deviation is in fact less than 1 nOhm/Ohm, consistent with array versus array comparisons at 8.5 mA. We can therefore conclude that neither of the subarrays have deviated much from nominal value, and that their discrepancies at high currents come mainly from one subarray starting to deviate slightly. This is further supported by S2, which explicitly shows that one subarray deviates from quantization before the other at low magnetic fields, consistent with expected differences in carrier density. The same behavior, that one subarray deviates from quantization before another, is then also expected when using high currents due to different breakdown current limits.

4) Many quantities must be italicized: fundamental constants, RK ,....

We thank the Reviewer for this. We have fixed these errors, mainly found in Supplementary Information figures.

5) When expressing quantities in tesla unit, replace magnetic field by magnetic induction!

We thank the Reviewer for this. We have changed all mentions to “magnetic flux density”

REVIEWER COMMENTS

Reviewer #1 (Remarks to the Author):

Third review 07/10

The authors ask for additional papers that would be appropriate as references. The following paper reports on optimal design of decade-value arrays using a method that minimizes the number of devices needed: Massimo Ortolano et al 2015 Metrologia 52 31.

P2, highlighted: "compares the mechanical power of a mass, moving at a certain velocity under the gravitational acceleration, to an electrical power determined by measured current I and voltage V ." This is not a correct summary of the Kibble Balance experiment, because the electrical power is virtual and obtained from the two separate parts of the experiment. The current only flows when the gravity force on the mass is balanced by the current in the stationary coil. The "power of a mass, moving at a certain velocity" is not encountered in any phase of that measurement because the mass is removed from the experimental apparatus for the velocity phase of the measurement.

P2, same paragraph as above: Your response letter indicates that you stand by the statement as written, "QHA devices with different resistances are also immensely useful for practical resistance metrology and will reduce uncertainties in calibration of a wide range of resistance values since they allow for one-to-one ratio comparisons, which are the most precise." I believe the direct one-to-one ratio is almost never used in precise ratio bridges where the most precise current sensitivity is required, such as at resistance values greater than 1000 ohms. In current comparator bridges both cryogenic and non-cryogenic, higher ratios allow current feedback to be applied to the side of the bridge with lower turns number, where noise sensitivity is reduced. When used in "practical resistance metrology" I think one-to-one measurements utilize either a modified Wheatstone bridge method or a current comparator with resistor substitution, which gives a one-to-one comparison in two steps.

The method employed in this paper to prove the subsection equality is a CCC bridge with equal turns (64/64) for the sub-arrays. This is fine, especially for the large currents used with these arrays, but no support is provided for what I disagree with, which is "QHA devices reduce uncertainties in calibration of a wide range of resistance values since they allow for one-to-one ratio comparisons, which are the most precise." Please correct this part of the quoted sentence.

Besides this, one-to-one comparisons would require matching of the array resistance to typically decade-value resistors, which is not addressed in this work. One earlier reference to such measurements is given by the authors in the reference suggested above, which states therein, "Resistance metrology would strongly benefit from the development of reliable QHARS having resistance values close to decadic values, because the calibration of artifact standards could thus be performed by 1:1 ratio bridges, which do not require ratio calibration, or even by substitution."

In the response letter it says, "The argument that agreement from comparison measurements can be due to pure coincidence can be used against all measurements, since one can never be 100 % sure due to finite uncertainties. However, the notion that subarrays would deviate from their ideal value in tandem is extremely improbable, since the agreement we observe is on the order of 10 nOhm, consistent across different bias currents and magnetic fields." There is no problem with the data in supporting the argument that these arrays are fully quantized. Your data is extremely thorough in support of the quantization. In the practical situation, where time is limited, a comparison of two sub-arrays under less-than-ideal conditions might not reveal possible errors that would be shown by R_{xx} measurements. If two sub-arrays differ by a certain amount, is it not possible that both are in error by a somewhat larger amount? This should be pointed out, in that QHR devices that are fabricated in close proximity on the same chip will tend to have very similar characteristics and will in some cases behave in a similar manner as they lose quantization.

The response says that additional guidelines are needed to allow wider use of QHARS devices (the original terminology) in resistance metrology. This is a very good point, and the metrology community needs to pursue such agreement. However, as you can see from the responses of

myself and Reviewer 3, it is not yet there. I suggest that you make this point more clearly in your conclusion, along with any other suggested extension of the current guidelines that is suited to arrays, which might help to convince the metrology community.

Sincerely,
Randolph E. Elmquist

Reviewer #3 (Remarks to the Author):

Report on revised paper (NCOMMS-21-51410B) entitled

“Exceptionally accurate large graphene quantum Hall arrays for the next SI”

From He *et al*

General comment

The authors have considered some remarks of the reviewers and made some changes. Some sentences, which could have been misleading were amended. It results that, now, the comparison of the two subarrays appears more clearly as a proposal of a future quantization test and not as a proof of accuracy. The fact remains that this quantization test requires a state-of-the-art CCC bridge, which reduces the autonomous character of the quantum standard for a routine application to the kilogram or the ampere realizations. However, it was demonstrated that it was possible to design parallel arrays (ref. 23) allowing the measurement of an upper value of the longitudinal resistance for the single Hall bars constituting the array. To my opinion, the ideal graphene parallel array, while benefitting from the superconducting links, will have to include R_{xx} measurements.

To demonstrate that the work brings new technology, the authors have included a few sentences and a new reference in the methods about the sample fabrication. This is interesting but the new revealed data are unfortunately not included in this paper. The introduction of some data about the technology (figure of three-layer resist, contact resistivity value achieved...) would have supported the technological contribution of the work and the broader interest required by Nature Communications papers. However, this work remains built on the previous technological achievements from NIST and co-workers reported in references 21, 31 and 22. On this subject, I consider that the work reported in ref 22 (mentioned in table 1), has to be described with more details since it reports on the “the first graphene QHR array (13 Hall bars in parallel) suitable as primary SI references”, as claimed by their authors. In the present state, the reader can miss it!

The novelty of the work, as it is written, remains the demonstration of the accuracy/reproducibility with very low measurement uncertainties of graphene arrays made of a large number of Hall bars connected using superconducting links. There too, the authors, seeking to enhance their own results, make a selection of previous reference works opened to criticism. In the state of the art of universality/reproducibility tests of arrays, selecting paper ref. 27 which reports a significant deviation of $(0.296 \pm 0.076) \times 10^{-9}$ and omitting paper (F. Schopfer et al, JAP 114, 064508, 2013), which reports an agreement of $(-0.002 \pm 0.032) \times 10^{-9}$ at zero current and $(0.003 \pm 0.04) \times 10^{-9}$ at 40 μ A current comforted by R_{xx} values measured with a few $\mu\Omega$ uncertainties (i.e. one hundred times lower those reported in He *et al* paper for the single Hall bar), on the pretext (in the rebuttal letter) that the second paper written by the same authors mentions Allan Variance measurements but does not explicitly report the data as in the first paper, is not justifiable : this paper was reviewed, accepted and has been commonly cited since 10 years. A fortiori, the authors mean that they would invalidate all the universality tests (Hartland et al, PRL 66, 969 (1991), Jeckelmann et al, PRB 55, 13124 (1997))

carried out in the 90s proving the agreement of silicon and GaAs quantum standards with

measurement uncertainties of 0.3×10^{-9} because they do not report on any Allan variance measurement. This makes no sense! Allan variance is of course a useful tool, now more currently used, but it does not replace an accurate measurement of R_{xx} !

To conclude, the paper reports on an important result for the metrological community. Without depreciating the paper results, the state-of-the-art should have been more completed and detailed. I keep reservations about the potential of the paper to meet a broader scientific community as expected for a publication in the Nature Communications journal. The technological improvements and progress in the devices fabrication, which support the results, are not clearly demonstrated in the paper itself. A clearer and more detailed link between measurement accuracy and technological improvements beyond the state of the art (notably compared to ref. 22) is required and would have interested a broader community.

Minor comments:

1) In methods, probably the thickness of the second polymer is 300 nm (and not 3000 nm)

2) In the supplementary, the term “error” is not correctly used.

- “The error denotes the standard deviation of the mean” should be replaced by “the uncertainty denotes the standard deviation of the mean”

- “The error bars are one standard error” should be replaced by “The errors bars are one standard deviations”.

Point-by-point reply

We would like to thank the two Reviewers for their additional questions. The Reviewers' comments are reproduced verbatim below and are written using normal font. The authors' answers are given using both bold and italic font. We have also formatted each question and answer into its own bullet point for better readability, including numbering like RX.X for ease of reference. The changes in the revised manuscript are marked with yellow highlights.

Reviewer #1 (Remarks to the Author):

- R1.1. Third review 07/10. The authors ask for additional papers that would be appropriate as references. The following paper reports on optimal design of decade-value arrays using a method that minimizes the number of devices needed: Massimo Ortolano et al 2015 Metrologia 52 31.

The suggested reference is already included in the main text (Ref. 19), and it shows that there exist design rules on how to achieve arbitrary resistance values for arrays with an optimal (low) number of devices. To create either low or high resistance arrays, the easiest method is still to simply connect devices either parallel or series, respectively. The algorithm in the paper comes into play when one wants to achieve exact resistance values such as 100 ohm, while minimizing the number of additional array elements needed to tune the exact resistance value.

For example, in a recent work which will be published soon (We will include a summary paper for the benefit of the Reviewers), we have designed a 100 ohm array using that very algorithm. In this recent work we required 151 individual Hall bars to reach exactly 100 ohm (within 0.2 nΩ/Ω deviation, per design). For comparison, a simple parallel connected array can reach around 100.5 ohm with 129 array elements.

In both our works, we want to push the limits of graphene arrays with as many Hall devices as possible, since more devices means more flexibility and range in achievable resistance values of a prospective array.

- R1.2. P2, highlighted: “compares the mechanical power of a mass, moving at a certain velocity under the gravitational acceleration, to an electrical power determined by measured current I and voltage V.” This is not a correct summary of the Kibble Balance experiment, because the electrical power is virtual and obtained from the two separate parts of the experiment. The current only flows when the gravity force on the mass is balanced by the current in the stationary coil. The “power of a mass, moving at a certain velocity” is not encountered in any phase of that measurement because the mass is removed from the experimental apparatus for the velocity phase of the measurement.

Following the Reviewer's comments, now we have reverted to a simpler statement of: “...which in a nutshell is an instrument which measures the weight of an object by balancing the gravitational force with a compensating electromagnetic force, which can be

defined in terms of the Planck constant h using the quantum Hall effect and the ac Josephson effect.”

- R1.3. P2, same paragraph as above: Your response letter indicates that you stand by the statement as written, “QHA devices with different resistances are also immensely useful for practical resistance metrology and will reduce uncertainties in calibration of a wide range of resistance values since they allow for one-to-one ratio comparisons, which are the most precise.” I believe the direct one-to-one ratio is almost never used in precise ratio bridges where the most precise current sensitivity is required, such as at resistance values greater than 1000 ohms. In current comparator bridges both cryogenic and non-cryogenic, higher ratios allow current feedback to be applied to the side of the bridge with lower turns number, where noise sensitivity is reduced. When used in “practical resistance metrology” I think one-to-one measurements utilize either a modified Wheatstone bridge method or a current comparator with resistor substitution, which gives a one-to-one comparison in two steps.

The benefit of one-to-one ratio measurements in the CCC-bridge is that one can directly check for coil error by switching the connection of the two standards under comparison. For instance, a measurement using Sample A/Coil A and Sample B/Coil B can be compared to a measurement using Sample A/Coil B and Sample B/Coil A. Furthermore, the unity ratio is less prone to ratio error coming from the addition of many coils needed to achieve the high ratio of ~129 necessary for 100 ohm versus single Hall bar device. We consistently observe that measuring one-to-one ratio produces the lowest noise, compared to larger ratios (e.g. array versus array has lower noise than array versus Hall bar).

In our laboratory we do commonly use direct one-to-one ratios in our CCC-system for routine calibrations. For instance, a transfer 100 ohm standard is commonly calibrated using another 100 ohm standard (itself calibrated against a graphene quantum standard). It is however uncommon to find direct one-to-one ratio between quantum standards and secondary standards in practical CCC-measurements because no primary quantum Hall standards exist with common decade resistance values. We believe our arrays presented herein have taken a big step towards this goal. As mentioned previously in the response to R1.1, we have already produced a precise array with exactly 100 ohm (see attached summary paper). Being able to achieve arbitrary resistance values using arrays allows us in the future to directly calibrate all secondary standards against a primary one, reducing the calibration chains and improving uncertainties. We have altered our statement to clarify this point (see also response below).

We agree with the Reviewer in that a simpler modified Wheatstone bridge could be used for one-to-one ratio measurements, instead of CCC. This highlights yet another benefit of arrays which allow for one-to-one measurements.

- R1.4. The method employed in this paper to prove the subsection equality is a CCC bridge with equal turns (64/64) for the sub-arrays. This is fine, especially for the large currents used with these arrays, but no support is provided for what I disagree with, which is “QHA devices reduce uncertainties in calibration of a wide range of resistance values since they allow for one-to-one ratio comparisons, which are the most precise.” Please correct this part of the quoted sentence.

Following the Reviewers comments, we have changed the statement to "...since they allow for direct comparison measurements between primary quantum standards and secondary standards, shortening the calibration chain".

Currently, only single Hall bar devices with ~ 12.9 kohm are used as a primary standard. This means that to cover the entire resistance range many intermediate secondary standards are required. For instance, we calibrate a 100 ohm standard using a quantum Hall standard, and then use that 100 ohm standard to calibrate other lower resistance standards, and so on. Each step in the calibration chain introduces uncertainties. However, by using arrays we can achieve arbitrary resistance values which allows us to directly calibrate all secondary standards against a primary one, reducing the calibration chains and improving uncertainties. We believe one-to-one ratio comparisons would be the simplest and most suitable (i.e. make arrays with common decade values).

- R1.5. Besides this, one-to-one comparisons would require matching of the array resistance to typically decade-value resistors, which is not addressed in this work. One earlier reference to such measurements is given by the authors in the reference suggested above, which states therein, "Resistance metrology would strongly benefit from the development of reliable QHARS having resistance values close to decadic values, because the calibration of artifact standards could thus be performed by 1:1 ratio bridges, which do not require ratio calibration, or even by substitution."

Our work lays the groundwork for precisely this goal, to achieve arbitrary resistance using primary array standards. We have now demonstrated the possibility of making large arrays with over 200 elements. The proof-of-concept is done, and it is possible to extend this work into other large arrays with for instance various decade resistance values.

As mentioned previously, we have already produced a precise array with the commonly used value of 100 ohm (300 array elements), which will be published shortly (see attached summary paper). This confirms our statement that, having demonstrated the principle, it is in now possible to produce arrays for one-to-one (or indeed arbitrary) ratio comparisons.

- R1.6. In the response letter it says, "The argument that agreement from comparison measurements can be due to pure coincidence can be used against all measurements, since one can never be 100 % sure due to finite uncertainties. However, the notion that subarrays would deviate from their ideal value in tandem is extremely improbable, since the agreement we observe is on the order of 10 nohm, consistent across different bias currents and magnetic fields." There is no problem with the data in supporting the argument that these arrays are fully quantized. Your data is extremely thorough in support of the quantization. In the practical situation, where time is limited, a comparison of two sub-arrays under less-than-ideal conditions might not reveal possible errors that would be shown by Rxx measurements. If two sub-arrays differ by a certain amount, is it not possible that both are in error by a somewhat larger amount? This should be pointed out, in that QHR devices that are fabricated in close proximity on the same chip will tend to have very similar characteristics and will in some cases behave in a similar manner as they lose quantization.

We stress that our array versus array comparison measurement is being proposed as the preferred test for routine measurements, with the understanding that rigorous checks - using all available quantization test methods - have been performed beforehand. This is the same requirement imposed on routine Rxx and contact resistance measurements. Since we have showed that our methods produce metrological standards with long-term stability (stable over years, DOI: 10.1088/1681-7575/ab2807), we expect that no significant deviation will suddenly occur. It is of course prudent to periodically perform rigorous tests, to ensure that the sample remains stable, and this is applicable for current single Hall bar standards also.

As we have pointed out previously, the expected random deviation of carrier density (and therefore resistance) across samples on the same chip should lead to orders of magnitude greater resistance deviations than the sub-part-per-billion agreement that we observe. We therefore assume a negligible chance that the agreement comes from anything other than quantized resistance of the array.

With that said, there are additional ways one could check the quantization for routine measurements. For instance, we have also brought up the fact that one could design future arrays to allow for both one-to-one ratio and a non-unity ratio comparison (for instance 100 ohm array divided into 3 parts, 25+25+50 ohm). The non-unity ratio could be used to detect parallel leakages. Furthermore, one can also design two arrays with identical resistances, but using different amount of individual hall bars for each. This can be achieved via redundant parallel and series connections. Single hall bars elements in one array can be replaced with four, by using two parallel connected sets of two serially connected Hall bars, without changing the total array resistance. This would mean that the current flowing through a single array Hall bar element would be different for the two arrays. A comparison measurement between the two would then be much more sensitive to potential quantization error, since if any of the Hall bars in an array deviate from perfect quantization, the different currents could lead to different resistance response. By combining these two aforementioned principles one could further alleviate any doubt regarding “accidental” quantization.

We have added these suggestions in the new summary section of the main text (see also response below).

- R1.7. The response says that additional guidelines are needed to allow wider use of QHARS devices (the original terminology) in resistance metrology. This is a very good point, and the metrology community needs to pursue such agreement. However, as you can see from the responses of myself and Reviewer 3, it is not yet there. I suggest that you make this point more clearly in your conclusion, along with any other suggested extension of the current guidelines that is suited to arrays, which might help to convince the metrology community.

We are happy that the Reviewer agree with us on the fact that the guideline need updating, and we look forward to the future discussions in the community regarding the practical implementation of array standards.

We feel that one of the main barriers currently is the fact that the wider community have not or cannot fabricate and test large quantum Hall arrays themselves. We hope that once our work has reached a wider audience, efforts to reproduce it will eventually lead to

interlaboratory comparisons of arrays, which is what is ultimately needed to establish graphene arrays as a primary resistance standard.

Our main suggestion is that subarray comparisons could be used for routine measurements using array standards, in lieu of R_{xx} and contact resistance. A related point is that the design of the array is important to allow for additional checks for quantization, refer to the response to R1.6 above for more details. With the rise of graphene array devices, using superconducting leads and effectively 2-probe quantum Hall measurements, some updates regarding recommended contact resistances and R_{xx} values could also be considered.

Following this discussion, we have extended the summary/conclusion section of the main text to:

“The device design of future arrays should also be taken into consideration. The subarray comparison measurements could be extended to allow for different ratio tests. For instance, a prospective 100 ohm array could be divided into 25+25+50 ohm parts. This would allow for non-unity ratio comparisons which serves as another quantization test and can also reveal potential errors like parallel leakages across the quantum Hall channel. Furthermore, one can also design two arrays with identical resistances, but using different amount of individual hall bars for each. This can be achieved via redundant parallel and series connections. Single hall bars elements in one array can be replaced by four elements, using two parallel connected sets of two serially connected Hall bars, without changing the total array resistance. This would mean that the current flowing through a single array Hall bar element would be different for the two arrays. A comparison measurement between the two would then be much more sensitive to potential quantization error, since if any of the Hall bars elements in either array deviate from perfect quantization, the different currents could lead to different resistance response. We foresee that these types of array-specific quantization tests will complement the existing single-hall bar tests in the future.

Embracing the use of array devices will allow for the QHE to be more intimately involved in the improvement of realizations of several key units, such as the ohm, ampere, and kilogram. We hope that our work will inspire further developments on this topic, and eventually lead to interlaboratory comparisons between different types of arrays, which is what is ultimately needed to establish graphene arrays as a primary resistance standard.”

Reviewer #3 (Remark to the Author):

- General comment

R3.1. The authors have considered some remarks of the reviewers and made some changes. Some sentences, which could have been misleading were amended. It results that, now, the comparison of the two subarrays appears more clearly as a proposal of a future quantization test and not as a proof of accuracy. The fact remains that this quantization test requires a state-of-the-art CCC bridge, which reduces the autonomous character of the quantum standard for a routine application to the kilogram or the ampere realizations. However, it was demonstrated that it was possible to design parallel arrays (ref. 23) allowing the measurement of an upper value of the longitudinal resistance for the single Hall bars constituting the array. To my opinion, the ideal graphene parallel array, while benefitting from the superconducting links, will have to include R_{xx} measurements.

We use our CCC-bridge to achieve the highest precision in primary resistance metrology, so it is natural that we would also use it to characterize of the array. The operation of Kibble balance and ampere realizations will take place in Metrology Institutes, where the majority have access to CCCs. With that said, the comparison measurement we are working with are not fundamentally tied to the CCC. Any resistance bridge could work, but the choice of setup will of course impact the measurement uncertainty. The benefit of arrays is that they have the possibility to achieve arbitrary resistances which allows for one-to-one comparisons to other standards using simpler designs like Wheatstone bridges.

We do not claim that R_{xx} should never be used, just that it is not always applicable. For instance, if one wished to achieve exact decade values of resistance one must use combinations of parallel and series connections, and the method in ref. 23 is not directly applicable. We therefore suggest the use of array comparisons for routine measurements in general.

We would like to reference our response to Reviewer #1 (R1.7). Reviewer #1 agrees with us on the fact that the guidelines need updating, and we look forward to the future discussions in the community regarding the practical implementation of array standards.

We reiterate that we feel that one of the main barriers currently is the fact that the wider community have not or cannot fabricate and test large quantum Hall arrays themselves. As stated in the new extended summary/conclusion section of the main text, we hope that once our work has reached a wider audience, efforts to reproduce it will eventually lead to interlaboratory comparisons of arrays, which is what is ultimately needed to establish graphene arrays as a primary resistance standard.

- R3.2. To demonstrate that the work brings new technology, the authors have included a few sentences and a new reference in the methods about the sample fabrication. This is interesting but the new revealed data are unfortunately not included in this paper. The

introduction of some data about the technology (figure of three-layer resist, contact resistivity value achieved...) would have supported the technological contribution of the work and the broader interest required by Nature Communications papers.

Our work on arrays represents the ultimate test of our latest graphene fabrication technologies, due to the high degree of precision required in every array element. We wish the check if the way we make quasi-edge contacts to graphene is truly scalable (<https://arxiv.org/abs/2206.03839>), and if the control of carrier density in epitaxial graphene by molecular doping (H. He et al., Nat. Commun., vol. 9, 2018) is truly homogeneous over large area.

The scope of this work is focused on the high precision measurement on large graphene arrays. We have already included all necessary details to reproduce our results in the Methods section, including detailed fabrication steps. The additional investigations into optimizing the fabrication methods fall outside the scope of this work, and we have therefore decided to tell that story in a separate publication, whose pre-print is referenced in the main text (<https://arxiv.org/abs/2206.03839>). The interested reader can have a look at that paper to read about the details like resist profile, distribution of contact resistivity, geometrical scaling behavior, reproducibility etc.

If the reviewer is familiar with microfabrication of devices, one key difference is that in this work on arrays we have extended the fabrication method to also include sputtered superconducting contacts using NbN, which is not trivial (in contrast to e-beam evaporated contacts described in our preprint). We have added some more additional clarifying details on this in the Methods section. The influence of contact resistivity can be inferred from our comparison measurements to be extremely low, since the effectively two-probe configuration of the array adds contact resistances directly to the quantized resistance. From our precision measurements, the influence of contact/lead resistances is only on the order of 20 nΩ ! We have added a clarifying sentence to highlight this fact in the summary section.

The reproducibility and reliability of our method is very high, and we have already repeated and improved upon the work in this paper, with an exact 100 ohm graphene array. This will be published shortly but we make it available to the Reviewers to build additional confidence in our results and corroborate some of our responses.

- R3.3. However, this work remains built on the previous technological achievements from NIST and co-workers reported in references 21, 31 and 22. On this subject, I consider that the work reported in ref 22 (mentioned in table 1), has to be described with more details since it reports on the “the first graphene QHR array (13 Hall bars in parallel) suitable as primary SI references”, as claimed by their authors. In the present state, the reader can miss it!

Our work builds upon many great previous achievements, one of such being NIST’s efforts which are already duly cited. Following the Reviewer’s comment, we single out the contribution of NIST to the field and have added the extra sentence to explicitly credit their work. In the section on device design and mention of superconducting leads it now reads “...following initial reports on primary standards using graphene arrays”.

However, the technology that we present, a large precise array capable of sustaining large currents, stands on a solid foundation on our own pioneering work on quantum resistance metrology with epigraphene since 2010. To name a few: our seminal work on quantum resistance metrology with epigraphene, which first reported the suitability of epitaxial graphene for quantum resistance metrology (A. Tzalenchuk et al., Nat. Nanotechnol., vol. 5, 2010); universality of QHE by comparing epigraphene and GaAs and redefinition of the SI (T. J. B. M. Janssen et al., New J. Phys., vol. 13, 2011); quality control of epigraphene material and quantum devices (T. Yager et al., Nano Lett., vol. 13, 2013, T. Yager et al., Carbon N. Y., vol. 87, 2015.). Our work also builds on our initial attempts to make large arrays with graphene (A. Lartsev et al., J. Appl. Phys., vol. 118, 2015), which were ultimately limited by our technology at the time (T. Yager et al., AIP Adv., vol. 5, 2015.).

Our experiences lead to two recent technological advances, which are subject to stringent tests in this work. To reiterate, we wish to check if the way we make quasi-edge contacts to graphene is truly scalable (<https://arxiv.org/abs/2206.03839>), and if the control of carrier density in epitaxial graphene by molecular doping (H. He et al., Nat. Commun., vol. 9, 2018) is truly homogeneous over large areas. Both technologies excel at reliably fabricating large amount of graphene devices, and we validate these technologies for the first time using one of the strictest tests: quantum resistance metrology.

- R3.4. The novelty of the work, as it is written, remains the demonstration of the accuracy/reproducibility with very low measurement uncertainties of graphene arrays made of a large number of Hall bars connected using superconducting links. There too, the authors, seeking to enhance their own results, make a selection of previous reference works opened to criticism. In the state of the art of universality/reproducibility tests of arrays, selecting paper ref. 27 which reports a significant deviation of $(0.296 \pm 0.076) \times 10^{-9}$ and omitting paper (F. Schopfer et al, JAP 114, 064508, 2013), which reports an agreement of $(-0.002 \pm 0.032) \times 10^{-9}$ at zero current and $(0.003 \pm 0.04) \times 10^{-9}$ at 40 μ A current comforted by R_{xx} values measured with a few μ ohm uncertainties (i.e. one hundred times lower those reported in He et al paper for the single Hall bar), on the pretext (in the rebuttal letter) that the second paper written by the same authors mentions Allan Variance measurements but does not explicitly report the data as in the first paper, is not justifiable : this paper was reviewed, accepted and has been commonly cited since 10 years. A fortiori, the authors mean that they would invalidate all the universality tests (Hartland et al, PRL 66, 969 (1991), Jeckelmann et al, PRB 55, 13124 (1997)) carried out in the 90s proving the agreement of silicon and GaAs quantum standards with measurement uncertainties of 0.3×10^{-9} because they do not report on any Allan variance measurement. This makes no sense! Allan variance is of course a useful tool, now more currently used, but it does not replace an accurate measurement of R_{xx} !

From our point of view, another key novelty is that we report an array of graphene quantum hall devices capable of sustaining large currents (~10 mA), giving graphene more possibilities to improve three key units of the new SI: the Ampere, the ohm and the kg.

We must emphasize that we are not aiming at discrediting older works. However, as metrology advances, new methods are developed and adopted. This is a common occurrence everywhere in research. We, and many in the community, strongly feel that Allan variance is of great importance to experimentally justify uncertainties, especially for

long series/averaging. In order to create fair comparisons, we therefore try to only include works which use Allan variance in Table 1.

Before moving on to the technical side of this discussion, we wish to first clarify that we have made no mention of Allan variance replacing R_{xx} . Neither R_{xx} nor Allan variance can replace each other since they are two different measurements that say entirely different things.

Allan variance is a very useful tool to experimentally verify the time stability and noise performance of a measurement campaign. It is used to check to what level of uncertainty white noise dominates, and the number of averages one can perform and still expect the uncertainty to scale down with square root of samples N . If no Allan variance is performed, one could theoretically average an arbitrary amount of measurement series and reach arbitrarily low uncertainties. Arbitrarily long averaging is not justified in practical experiments since there are other uncertainty sources (e.g. drift) which can dominate at longer time scales. That is why Allan variance is useful and sees increasing use in metrology. We have deliberately chosen to limit our stated uncertainties to experimentally verified Allan variance, and not to the much lower calculated values.

With that being said, we have included the suggested reference (F. Schopfer et al, JAP 114, 064508, 2013) in the parts where we mention other universality tests and their reported uncertainties. The reference can fit there since those reported results do not necessarily abide by the experimental limit of Allan variance either. We have previously already included another related paper from the same authors, from 2007, using Wheatstone bridge. There they claimed around $0.07 n\Omega/\Omega$, with supporting Allan variance. We have now included a new sentence after the mention of their previous results: "...and even $0.032 n\Omega/\Omega$ ", with a reference to the 2013 paper suggested by the Reviewer.

- R3.5. To conclude, the paper reports on an important result for the metrological community. Without depreciating the paper results, the state-of-the-art should have been more completed and detailed. I keep reservations about the potential of the paper to meet a broader scientific community as expected for a publication in the Nature Communications journal. The technological improvements and progress in the devices fabrication, which support the results, are not clearly demonstrated in the paper itself. A clearer and more detailed link between measurement accuracy and technological improvements beyond the state of the art (notably compared to ref. 22) is required and would have interested a broader community.

We are happy that the Reviewer recognize that our results represent the state-of-the-art of arrays. In accordance with the Reviewer, we have implemented further changes in the Methods section, to further clarify the device fabrication method. We have also added an additional sentence in the summary/conclusion section of the main text which explicitly states that this method is key to ensuring low contact resistances through the whole array.

"The reliable fabrication of such a precise array is dependent on key enabling technologies such as homogenous molecular doping and creation of low contact resistance superconducting leads on all array elements (the influence of contact resistance is estimated to be $< 20 n\Omega$)."

The description of the method is complete and already includes all necessary details needed to reproduce these results in full. The unprecedented size and precision of the arrays are already very strong supporting argument for our fabrication method. Further details on the development of the fabrication technique lie outside the scope of this work, and can be found in the aforementioned separate publication (<https://arxiv.org/abs/2206.03839>).

We wish to stress yet again that our hope is that this work will stimulate more research efforts on graphene array in the wider community. This will hopefully lead to interlaboratory comparisons of arrays, which is what is ultimately needed to establish graphene arrays as a primary resistance standard. We have added this sentiment to the summary/conclusion section. Furthermore, we also believe that these results are of even broader interest, not only in the metrology community due to the improvement to standards, but in general for people who are interested in scaled up production of graphene electronics.

- Minor comments:
 - 1) In methods, probably the thickness of the second polymer is 300 nm (and not 3000 nm)
 - 2) In the supplementary, the term “error” is not correctly used.
 - “The error denotes the standard deviation of the mean” should be replaced by “the uncertainty denotes the standard deviation of the mean”
 - “The error bars are one standard error” should be replaced by “The errors bars are one standard deviations”.

We thank the Reviewer for the minor comments above. We have fixed all of them.

REVIEWERS' COMMENTS

Reviewer #1 (Remarks to the Author):

Thank you for addressing the items that were brought up at the onset and later in the review process. My most significant concerns have been addressed.